A taxonomic review of the Late Jurassic eucryptodiran turtles from the Jura Mountains (Switzerland and France)

Anquetin Jérémy 1 2 j.anquetin@gmail.com
Püntener Christian 1
Billon-Bruyat Jean-Paul 1
1 Section d’archéologie et paléontologie, Office de la culture, République et Canton du Jura , Porrentruy , Switzerland
2 UMR CNRS 7207 MNHN UPMC, Muséum national d’histoire naturelle , Paris , France
Schneider Richard
Electronic publication date: 2014 May 13
Publication date: 2014
Volume: 2
Electronic Location ID: e369
Received 2014 Mar 14; Accepted 2014 Apr 11
Copyright: © 2014 Anquetin et al.
Copyright year: 2014
Copyright holder: Anquetin et al.
License: This is an open access article distributed under the terms of the Creative Commons Attribution License, which permits unrestricted use, distribution, reproduction and adaptation in any medium and for any purpose provided that it is properly attributed. For attribution, the original author(s), title, publication source (PeerJ) and either DOI or URL of the article must be cited.
License URL: https://creativecommons.org/licenses/by/4.0/

Keywords: Plesiochelys, Craspedochelys, Tropidemys, Thalassemys, Plesiochelyidae, Thalassemydidae, Testudines, Kimmeridgian, Tithonian

Funding: Simone and Cino del Duca Foundation Federal Roads Office Republic and Canton of Jura Jérémy Anquetin’s visits to the NMS and MH in 2010 were funded as part of a postdoctoral grant from the Simone and Cino del Duca Foundation (2008 Foundation Prize awarded to Philippe Janvier, MNHN, Paris). The PAL A16 team (Section d’archéologie et paléontologie) is funded by the Federal Roads Office (FEDRO, 95%) and the Republic and Canton of Jura (RCJU, 5%). The funders had no role in study design, data collection and analysis, decision to publish, or preparation of the manuscript.

==============================
Background. Eucryptodiran turtles from the Late Jurassic (mainly Kimmeridgian) deposits of the Jura Mountains (Switzerland and France) are among the earliest named species traditionally referred to the Plesiochelyidae, Thalassemydidae, and Eurysternidae. As such, they are a reference for the study of Late Jurassic eucryptodires at the European scale. Fifteen species and four genera have been typified based on material from the Late Jurassic of the Jura Mountains. In the past 50 years, diverging taxonomic reassessments have been proposed for these turtles with little agreement in sight. In addition, there has been a shift of focus from shell to cranial anatomy in the past forty years, although most of these species are only represented by shell material. As a result, the taxonomic status of many of these 15 species remains ambiguous, which prevents comprehensive comparison of Late Jurassic turtle assemblages throughout Europe and hinders description of new discoveries, such as the new assemblage recently unearthed in the vicinity of Porrentruy, Switzerland.

Methods. An exhaustive reassessment of the available material provides new insights into the comparative anatomy of these turtles. The taxonomic status of each of the 15 species typified based on material from the Late Jurassic of the Jura Mountains is evaluated. New diagnoses and general descriptions are provided for each valid taxon.

Results. Six out of the 15 available species names are recognized as valid: Plesiochelys etalloni, Craspedochelys picteti, Craspedochelys jaccardi, Tropidemys langii, Thalassemys hugii, and ‘Thalassemys’ moseri. The intraspecific variability of the shell of P. etalloni is discussed based on a sample of about 30 relatively complete specimens from Solothurn, Switzerland. New characters are proposed to differentiate P. etalloni, C. picteti, and C. jaccardi, therefore rejecting the previously proposed synonymy of these forms. Based partly on previously undescribed specimens, the plastral morphology of Th. hugii is redescribed. The presence of lateral plastral fontanelles is notably revealed in this species, which calls into question the traditional definitions of the Thalassemydidae and Eurysternidae. Based on these new data, Eurysternum ignoratum is considered a junior synonym of Th. hugii. The Eurysternidae are therefore only represented by Solnhofia parsonsi in the Late Jurassic of the Jura Mountains. Finally, ‘Th.’ moseri is recognized as a valid species, although a referral to the genus Thalassemys is refuted.

Introduction

From 2000 to 2011, controlled excavations along the future course of the A16 Transjurane highway have opened an unprecedented window into the late Kimmeridgian of the Jura Mountains, in the vicinity of Porrentruy (Canton of Jura, NW Switzerland; Fig. 1). The mission of the PAL A16 team (Section d’archéologie et paléontologie, Office de la culture, République et Canton du Jura, Switzerland) was first to document the geology and paleontology of intersected sedimentary rocks. This notably led to the discovery of a rich and diverse Mesozoic coastal marine vertebrate fauna, including fishes, turtles, crocodilians, and pterosaurs, and several extensive dinosaur track-bearing sites (e.g., Marty & Hug, 2003; Billon-Bruyat, 2005; Marty et al., 2007; Marty & Billon-Bruyat, 2009). The PAL A16 team is now entering the second phase of its mission: the scientific study of this rich material. Turtles are among the most abundant vertebrates discovered during the excavations. Up to now, the PAL A16 Mesozoic turtle collection includes about 80 shells (more than 50 of which are already prepared), five crania, four mandibles and thousands of isolated remains. Preliminary investigations reveal that this turtle assemblage is taxonomically diverse. A recent study focused on the species Tropidemys langii Rütimeyer, 1873 and described new, articulated material collected by the PAL A16 team that considerably improved our knowledge of this characteristic Late Jurassic plesiochelyid turtle (Püntener et al., 2014). The PAL A16 turtle assemblage also includes several taxa that can be provisionally referred to the traditional families Plesiochelyidae Baur, 1888, Thalassemydidae Zittel, 1889, and Eurysternidae Dollo, 1886. However, the definitions and diagnoses of these families are rather confused (Joyce, 2003; Joyce, 2007; Anquetin, Deschamps & Claude, 2014; Anquetin & Joyce, in press).

Figure 1 Map showing the location of the Late Jurassic turtle sites throughout the Jura Mountains (Switzerland and France).

Numerous eucryptodiran turtles have been collected from the Kimmeridgian of the Jura Mountains since the early nineteenth century, notably from the famous Solothurn Turtle Limestone (Canton of Solothurn, Switzerland; e.g., Rütimeyer, 1873; Bräm, 1965; Meyer & Thüring, 2009) and from the vicinity of Porrentruy (e.g., Marty & Billon-Bruyat, 2009; Püntener et al., 2014). A total of five localities have produced significant turtle material (Fig. 1). Fifteen species, including Plesiochelys etalloni (Pictet & Humbert, 1857), P. solodurensis Rütimeyer, 1873, Craspedochelys picteti Rütimeyer, 1873, C. jaccardi (Pictet, 1860), Tropidemys langii, Thalassemys hugii Rütimeyer, 1873, and Th. moseri Bräm, 1965, have been typified based on specimens from the Late Jurassic of the Jura Mountains. The PAL A16 Mesozoic turtle collection must therefore be directly compared to these early finds, but there is currently no proper agreement upon their taxonomy. In particular, previous authors disagreed on the number of species represented in Solothurn (see Previous Work, below). This situation prevents a detailed interpretation of the PAL A16 turtle assemblage.

Plesiochelys solodurensis, C. picteti, Tr. langii, and Th. hugii are the type species of their respective genera. Similarly, Plesiochelys Rütimeyer, 1873 and Thalassemys Rütimeyer, 1859a are the type genera of plesiochelyids and thalassemydids, respectively. Late Jurassic turtles from the Jura Mountains are therefore of major importance for the taxonomy of basal eucryptodires. However, since Bräm (1965), no author has properly reassessed the shell morphology of these forms. The purpose of the present contribution is to review the taxonomy of the 15 turtle species typified based on fossil specimens from the Late Jurassic of the Jura Mountains. This study is tightly linked to a recent paper in which we announced the rediscovery of the holotype material of P. etalloni (Anquetin, Deschamps & Claude, 2014). The type material of each of these 15 species has been carefully scrutinized in order to evaluate the taxonomies proposed by recent authors (e.g., Gaffney, 1975a; Lapparent de Broin, Lange-Badré & Dutrieux, 1996). Many additional specimens from Solothurn have also been studied first-hand as part of the present work (Table S1). This reassessment is an essential first step toward a broader revision of the plesiochelyids and thalassemydids at the European scale and will serve as a solid reference for the interpretation of new discoveries, most notably the rich material recently unearthed by the PAL A16 team in the vicinity of Porrentruy.

Previous Work

Bräm (1965) related the history of the Solothurn turtle collection in details. This collection, whose origins go back to the years 1820 and 1830, is tied to the fate of Professor FJ Hugi, keen naturalist and alpinist, who established the “Naturforschenden Gesellschaft Solothurn” (Society of Natural History of Solothurn) in 1823 (Lienhard, 2008). Being the first to recognize the presence of turtle remains in the Solothurn quarries, he gave a lecture on his fossil turtle collection in 1824 during a meeting of the “Naturforschenden Gesellschaft Solothurn” and even sent information and specimens to G Cuvier, who figured a turtle skull (NMS 134) and other specimens from Solothurn in the second edition of his Recherches sur les ossemens fossiles (Cuvier, 1824: 227–232; Bräm, 1965; Gaffney, 1975a). In 1825, FJ Hugi sold his private collection to the city and was appointed as first director of the newly created city museum (Meyer & Thüring, 2009). A few decades later, Professor F Lang, the successor of FJ Hugi as the head of the Solothurn museum, appointed Dr L Rütimeyer to study the huge turtle collection in question. Lang & Rütimeyer (1867) published a first account on the geology of the Solothurn quarries that contained a monograph on the specimens referable to Platychelys oberndorferi Wagner, 1853, a species originally defined based on a specimen from the Tithonian of Kelheim, Germany. Unaware of the publication of Wagner (1853), Rütimeyer (1859a) originally described the Solothurn specimens as a new genus called Helemys. Interestingly, several authoritative references (e.g., Lydekker, 1889; Kuhn, 1964; Bräm, 1965) mentioned the species Helemys serrata (Rütimeyer, 1859a), but it is unclear when the epithet serrata was actually associated with this genus name. Rütimeyer (1859a), Rütimeyer (1859b), Rütimeyer (1873), Lang & Rütimeyer (1867), and Maack (1869) all referred to Helemys without a specific epithet. This should ultimately be investigated. Platychelys oberndorferi is a panpleurodiran turtle and we will not expand further on this taxon in the present study, which is restricted to eucryptodires. All the other then known turtles from Solothurn were described in Rütimeyer (1873).

At the time when Rütimeyer was starting to work on the Solothurn material, the Swiss paleontologist F-J Pictet described two new turtles from the Late Jurassic of the Jura Mountains: Emys etalloni Pictet & Humbert, 1857 from the vicinity of Moirans-en-Montagne (Department of Jura, France) and Emys jaccardi Pictet, 1860 from Les Hauts-Geneveys (Canton of Neuchâtel, Swiss Jura; Fig. 1). Rütimeyer (1873) subsequently referred these two species to his newly created genus Plesiochelys. Surprisingly, despite being the reference for the application of the species names etalloni and jaccardi, this material received relatively little attention since Pictet’s time.

Bräm (1965) provided a detailed reassessment of the Solothurn turtle fauna, including specimens discovered after the work of Rütimeyer. This contribution remains a major reference today. Each of the 13 species recognized by Rütimeyer (1873) was evaluated; this count excludes Platychelys oberndorferi, which is both non-controversial and not typified based on material from the Jura Mountains (see above). The validity of eight species was confirmed and two new species were created. Table 1 summarizes the taxonomy proposed by different authors discussed herein. According to Bräm (1965), there is only one species of Craspedochelys and Tropidemys in Solothurn, instead of three in each genus as proposed by Rütimeyer (1873). Bräm (1965) recognized two species of Thalassemys in Solothurn: Th. hugii and Th. moseri, a new species. In addition, Bräm (1965) erected a new species, Eurysternum ignoratum, for some of the remains previously assigned to Thalassemys by Rütimeyer (1873). The two authors were more or less on the same line concerning the genus Plesiochelys as Bräm (1965) still recognized the presence of four species out of the five originally described in Solothurn (P. etalloni, P. jaccardi, P. solodurensis and P. sanctaeverenae).

Table 1 Summary of the various taxonomies proposed for the eucryptodiran turtles typified based on material from the Late Jurassic of the Jura Mountains since Rütimeyer (1873).

Blank cells represent synonymies; n-dash indicates that the taxon was not considered in the concerned study.

Rütimeyer (1873)	Bräm (1965)	Gaffney (1975a)	Lapparent de Broin, Lange-Badré & Dutrieux (1996)	This study	
P. Etalloni	P. etalloni	P. etalloni	P. etalloni	P. etalloni	
P. solodurensis	P. solodurensis		P. solodurensis		
P. Langii	(partial)		–		
P. Sanctae Verenae	P. sanctaeverenae		–		
P. Jaccardi	P. jaccardi		C. jaccardi	C. jaccardi	
C. Picteti	C. picteti		C. picteti	C. picteti	
C. crassa		–	–		
Tr. Langii	Tr. langii	–	Tr. langii	Tr. langii	
C. plana		–	–		
Tr. expansa		–	–		
Tr. gibba		–	–		
Th. Hugii	Th. hugii	–	Th. hugii	Th. hugii	
Th. Gresslyi		–	–		
–	E. ignoratum	–	–		
–	Th. moseri	–	Ref. to P. solodurensis	‘Th.’ moseri	
Notes.

C. Craspedochelys

E. Eurysternum

P. Plesiochelys

Th. Thalassemys

Tr. Tropidemys

Ten years later, ES Gaffney developed an interest for Late Jurassic turtles from Europe. In contrast to previous workers, he focused more specifically on cranial material, often considering that the turtle shell was subject to too many individual variations to be heavily relied upon for systematic purposes (e.g., Gaffney, 1972; Gaffney, 1975a). In the Late Jurassic of the Jura Mountains, only plesiochelyids have produced significant cranial material. Studying the material from Solothurn and Glovelier (Canton of Jura, Switzerland; Fig. 1), Gaffney (1975a) reached the conclusion that all available skulls should be assigned to a single species. In parallel, he rejected all the shell-based arguments proposed by Bräm (1965) to support the distinction between the various Plesiochelys and Craspedochelys species. He notably synonymized P. solodurensis, P. jaccardi, P. sanctaeverenae and C. picteti with P. etalloni (Pictet & Humbert, 1857).

In contrast to previous authors, Antunes, Becquart & Broin (1988) assigned Emys jaccardi Pictet, 1860 to the genus Craspedochelys Rütimeyer, 1873, creating the new combination Craspedochelys jaccardi (Pictet, 1860). They also suggested that European plesiochelyids and thalassemydids should be revised, as neither Bräm’s (1965) nor Gaffney’s (1975a) taxonomies were satisfactory. Although they did not propose an extensive revision of these groups, Lapparent de Broin, Lange-Badré & Dutrieux (1996) provided some ideas regarding their taxonomy and possible relationships (Table 1). They recognized two valid species in the genus Plesiochelys, P. etalloni and P. solodurensis. They considered that only one species of Plesiochelys was present in Solothurn, P. solodurensis, and that P. etalloni was closely related but different. Craspedochelys jaccardi and C. picteti were also considered as valid. Finally, thalassemydids were restricted to Thalassemys hugii, and Th. moseri was synonymized with P. solodurensis.

Systematic Paleontology

TESTUDINES Batsch, 1788 EUCRYPTODIRA Gaffney, 1975c PLESIOCHELYIDAE Baur, 1888

Plesiochelys Rütimeyer, 1873

1873 Plesiochelys. Rütimeyer [new genus]

Type species. Plesiochelys solodurensis Rütimeyer, 1873.

Revised diagnosis. Type genus of the Plesiochelyidae, which are traditionally defined as having three cervical scales and a completely ossified carapace. Differing from Craspedochelys in: carapace more elongated and oval; deeper nuchal notch usually extending laterally up to the middle of peripheral 1; lower length/width ratio of costal bones (3.1–3.6, as opposed to 4.3–4.8 or more for the fourth costal in Craspedochelys); relatively long plastron (about 85–90% of carapace length); hyoplastron longer than wide. Differing from Tropidemys in: absence of neural keel; elongated neurals; wider vertebral scales.

Remarks. Closely following the original definition of Rütimeyer (1873), Bräm (1965) mainly diagnosed Plesiochelys based on the following characters: carapace oval in outline, usually longer than wide; nuchal notch extending laterally up to the middle of the first peripheral; plastron large and oval in outline; anterior lobe with bulbous epiplastral processes; posterior lobe rounded, occasionally slightly notched; entoplastron wide and often shield-shaped; hyoplastra longer than hypoplastra; central plastral fontanelle present or absent; pelvic girdle connected to plastron by means of the prepubic process. Gaffney (1975a: 6) proposed a slightly updated diagnosis, which by his own opinion “does not serve as a satisfactory set of identifying criteria”.

Proposing a complete diagnosis for the genus Plesiochelys is indeed relatively complicated as several Late Jurassic forms from Europe, notably from Germany, France and Portugal (e.g., Antunes, Becquart & Broin, 1988; Lapparent de Broin, 2001; Karl et al., 2007), are in need of serious reconsideration. Considering only Plesiochelys and not other potentially synonymous genera, a score of species are typified based on European material (Kuhn, 1964). As it stands, only two of these species are currently sufficiently known: P. etalloni (sensu Anquetin, Deschamps & Claude, 2014) and P. planiceps (Owen, 1842) from the Tithonian of southern England (Gaffney, 1975a). The latter is known by a single, isolated cranium with associated mandible. The above revised diagnosis is a first step, which will be refined in the future as the understanding of the taxonomy of Late Jurassic European turtles improves.

Plesiochelys solodurensis Rütimeyer, 1873

1873 Plesiochelys solodurensis. Rütimeyer [new species]

1975a Plesiochelys etalloni. Gaffney [subjective synonymy]

2014 Plesiochelys etalloni. Anquetin, Deschamps & Claude [subjective synonymy]

Taxonomic assessment. Invalid name, subjective synonym of Plesiochelys etalloni (Pictet & Humbert, 1857).

Type material. NMS 59, a distorted sub-complete shell. Lectotype designated by Bräm (1965: 81).

Type horizon and locality. Solothurn Turtle Limestone, uppermost member of the Reuchenette Formation (Kimmeridgian, Late Jurassic), vicinity of Solothurn, Canton of Solothurn, Switzerland (Fig. 1).

Illustrations of type. Rütimeyer (1873: plate XII, Figs. 1 and 2); Figs. 2E–2H.

Figure 2 Plesiochelys etalloni.

Holotype of P. etalloni, MAJ 2005-11-1: (A) photograph of the carapace; (B) interpretative drawing of the carapace; (C) photograph of the plastron; (D) interpretative drawing of the plastron. Lectotype of P. solodurensis, NMS 59: (E) photograph of the carapace; (F) interpretative drawing of the carapace; (G) photograph of the plastron; (H) interpretative drawing of the plastron. Lectotype of P. sanctaeverenae, NMS 118: (I) photograph of the carapace; (J) interpretative drawing of the carapace. Lectotype of P. langii, NMS 123: (K) photograph of the carapace; (L) interpretative drawing of the carapace. Bones are white; stripped lines indicate internal bone layers; green solid lines indicate scales sulci; matrix is gray. Abbreviations: eb, epiplastral bulb; n, neural; *, intermediate element (see text).

Remarks. Rütimeyer (1873) and Bräm (1965) referred most of the Plesiochelys material from Solothurn either to P. solodurensis or P. etalloni. The main difference they recognized between the two species was the presence of a central plastral fontanelle in P. etalloni. Bräm (1965: 60–62) even concluded that the two species were very similar, to the point of being hardly differentiable if hyo- and hypoplastron were not preserved. Considering the fact that the retention of small shell fontanelles is intraspecifically variable in some extant species (e.g., Pelomedusa subrufa, Graptemys barbouri, and Macroclemys temminckii; see Pritchard, 2008), Gaffney (1975a) assumed this was also the case in Plesiochelys and referred P. solodurensis, P. jaccardi and P. etalloni to the same species. Lapparent de Broin, Lange-Badré & Dutrieux (1996) also considered the persistence of a small central plastral fontanelle in adults as an intraspecific variation of P. solodurensis, which they considered as a distinct species. Based on an extensive review of the relevant material, we reached a conclusion similar to that of Gaffney (1975a) and Lapparent de Broin, Lange-Badré & Dutrieux (1996), although we ultimately disagree on the delimitation and inclusiveness of Plesiochelys species (Anquetin, Deschamps & Claude, 2014; see also Table 1 and Discussion).

Plesiochelys etalloni (Pictet & Humbert, 1857)

1857 Emys etalloni. Pictet & Humbert [new species]

1873 Plesiochelys Etalloni. Rütimeyer [new combination]

Taxonomic assessment. Valid name.

Synonymy. Plesiochelys solodurensis Rütimeyer, 1873, Plesiochelys sanctaeverenae Rütimeyer, 1873, and Plesiochelys langii Rütimeyer, 1873.

Type material. MAJ 2005-11-1, a shell missing a large part of the carapace medially. Holotype by monotypy.

Type horizon and locality. “Forêt de Lect” near Moirans-en-Montagne (Department of Jura, France; Fig. 1), possibly early Tithonian (but see Anquetin, Deschamps & Claude, 2014), Late Jurassic.

Illustrations of type. Pictet & Humbert (1857: plates I-III); Anquetin, Deschamps & Claude (2014: Figs. 1 and 2, S2 and S3); Figs. 2A–2D.

Referred specimens. See Bräm (1965): specimens referred to P. etalloni, P. solodurensis, and P. sanctaeverenae. For cranial material, see Gaffney (1975a).

Revised diagnosis. See Anquetin, Deschamps & Claude (2014).

Remarks. Lost for more than 150 years, the holotype of P. etalloni has been recently relocated. Thanks to this rediscovery, the taxonomy of this species was revised (Anquetin, Deschamps & Claude, 2014). For the purpose of the present study, we have reassessed a great number of specimens from the Late Jurassic of the Jura Mountains. In contrast to Gaffney (1975a), we reached the conclusion that Craspedochelys picteti and C. jaccardi are not synonyms of P. etalloni (see below).

Plesiochelys sanctaeverenae Rütimeyer, 1873

1873 Plesiochelys Sanctae Verenae. Rütimeyer [new species]

1975a Plesiochelys etalloni. Gaffney [subjective synonymy]

2014 Plesiochelys etalloni. Anquetin, Deschamps & Claude [subjective synonymy]

Taxonomic assessment. Invalid name, subjective synonym of Plesiochelys etalloni (Pictet & Humbert, 1857).

Type material. NMS 118, a large carapace missing both lateral parts. Lectotype designated by Bräm (1965: 126).

Type horizon and locality. Solothurn Turtle Limestone, uppermost member of the Reuchenette Formation (Kimmeridgian, Late Jurassic), vicinity of Solothurn, Canton of Solothurn, Switzerland (Fig. 1).

Illustrations of type. Rütimeyer (1873: plate XIII); Figs. 2I and 2J.

Remarks. According to Bräm (1965), P. sanctaeverenae differs from P. solodurensis and P. etalloni by its greater size (up to 550 mm), a more elongate carapace, a well-developed nuchal notch, and well-developed sulci. However, Bräm (1965: 127) himself admitted that the morphology of NMS 118 (lectotype of P. sanctaeverenae) was in fact very similar to that of the largest specimens he otherwise referred to P. etalloni or P. solodurensis. Gaffney (1975a) attributed these minor differences to individual variations and synonymized P. sanctaeverenae with P. etalloni. According to Lapparent de Broin, Lange-Badré & Dutrieux (1996), only one species of Plesiochelys (P. solodurensis, not P. etalloni) is present in Solothurn, which implies that they considered P. sanctaeverenae as a synonym of P. solodurensis, although they did not make that clear in their paper. A recent review of the relevant material confirmed that it is impossible to differentiate NMS 118 from other specimens referred to P. etalloni (Anquetin, Deschamps & Claude, 2014).

Plesiochelys langii Rütimeyer, 1873

1873 Plesiochelys Langii. Rütimeyer [new species]

1965 Plesiochelys solodurensis. Bräm [subjective synonymy]

2014 Plesiochelys etalloni. Anquetin, Deschamps & Claude [subjective synonymy]

Taxonomic assessment. Invalid name, subjective synonym of Plesiochelys etalloni (Pictet & Humbert, 1857).

Type material. NMS 123, a sub-complete carapace missing the right and posterior margins. Herein designated as lectotype (see Remarks, below). NMS 126, a shell heavily encrusted with pyritic mineralizations (paralectotype).

Type horizon and locality. Solothurn Turtle Limestone, uppermost member of the Reuchenette Formation (Kimmeridgian, Late Jurassic), vicinity of Solothurn, Canton of Solothurn, Switzerland (Fig. 1).

Illustrations of type. Rütimeyer (1873: plate VI, Figs. 1 and 2); Figs. 2K and 2L.

Remarks. Rütimeyer (1873) erected Plesiochelys langii based on three specimens, which together form the original syntype series: NMS 123, NMS 124 and NMS 126. He primarily differentiated P. langii based on a circular carapace outline and unusually wide peripherals forming alternating projections with costals. Bräm (1965) attributed these features to individual variation or postmortem deformation and synonymized P. langii with P. solodurensis. A recent review of the available material confirmed that NMS 123 and NMS 126 do not significantly differ from other specimens referred to P. etalloni, notably NMS 59 (lectotype of P. solodurensis) and MAJ 2005-11-1 (holotype of P. etalloni). Therefore, P. langii was synonymized with P. etalloni (Anquetin, Deschamps & Claude, 2014). As pointed out by Bräm (1965), NMS 124 clearly belongs to a different species (see below). In order to avoid potential future issues with the taxonomic status of P. langii, NMS 123, the main specimen described by Rütimeyer (1873), is herein designated as the lectotype of this species.

NMS 124 was initially described by Rütimeyer (1873) as a juvenile individual of P. langii. Bräm (1965) first recognized that this specimen belonged to a different taxon: the vertebrals are reduced in width and costo-peripheral fontanelles are present. However, the exact opinion of Bräm (1965) upon the correct attribution of this specimen remains somewhat confusing. At first, he declared that the specimen should be attributed to Thalassemys (ibid.: 29). Then, he seemed to hesitate between a referral to Thalassemys and one to Eurysternum ignoratum, finally concluding that, given the great correspondence between NMS 124 and NMS 5 (the type of E. ignoratum), the latter identification was more likely (ibid.: 168). NMS 124 is herein referred to Thalassemys hugii (see below).

Craspedochelys Rütimeyer, 1873

1873 Craspedochelys. Rütimeyer [new genus]

1975a Plesiochelys. Gaffney [subjective synonymy]

Type species. Craspedochelys picteti Rütimeyer, 1873.

Revised diagnosis. Form traditionally referred to the Plesiochelyidae based on the presence of three cervical scales and a completely ossified carapace. Differing from Plesiochelys in: broad, more rounded carapace, usually as wide as long (as preserved); shallower nuchal notch usually restricted to nuchal plate; higher length/width ratio of costal bones (4.3–4.8 or more, as opposed to 3.1–3.6 for the fourth costal in Plesiochelys); hyoplastron proportionally wider (even wider than long in C. jaccardi). Differing from Tropidemys in: absence of neural keel; elongated neurals; wider vertebral scales.

Remarks. According to Bräm (1965), Craspedochelys is monospecific and the diagnosis he provided is therefore restricted to C. picteti: carapace as wide as long and shaped like a heraldic shield; anterior carapace rim almost straight up to third peripheral, then bending almost at right angle toward the rear; weak nuchal notch; free first thoracic rib, articulated neither to first costal nor to second thoracic rib; second thoracic rib stronger than following ones and connected only to second thoracic vertebra. Gaffney (1975a) tentatively synonymized Craspedochelys with Plesiochelys, explaining differences in shell outline and development of the nuchal notch by postmortem compression and individual variation, respectively. However, he concluded that the condition of the first and second thoracic ribs may prove to be consistent when more specimens are known. Subsequent studies tended to re-establish a distinction between Craspedochelys and Plesiochelys, based mostly on shell shape criteria (Antunes, Becquart & Broin, 1988; Lapparent de Broin, Lange-Badré & Dutrieux, 1996).

Morphologically, Craspedochelys and Plesiochelys are relatively close. However, as already noted by Lapparent de Broin, Lange-Badré & Dutrieux (1996), the available material from the Jura Mountains clearly reveals two morphotypes: Plesiochelys has a more elongate carapace and a relatively long plastron, whereas Craspedochelys has a broader, more rounded carapace (more or less as wide as long, as preserved) and a shorter plastron (only known in C. jaccardi). These differences cannot be explained by postmortem deformation alone (see Discussion). In the course of the present study, we have also identified a set of characters related to the proportions of various shell elements that differentiate Craspedochelys from Plesiochelys (see Discussion).

Craspedochelys picteti Rütimeyer, 1873

1873 Craspedochelys Picteti. Rütimeyer [new species]

1975a Plesiochelys etalloni. Gaffney [subjective synonymy]

1988 Craspedochelys jaccardi. Antunes, Becquart & Broin [subjective synonymy]

Taxonomic assessment. Valid name.

Synonymy. Craspedochelys crassa Rütimeyer, 1873.

Type material. NMS 129, anterior half of a shell with plastron poorly preserved and right part of the carapace missing. Holotype by monotypy (Bräm, 1965: 137).

Type horizon and locality. Solothurn Turtle Limestone, uppermost member of the Reuchenette Formation (Kimmeridgian, Late Jurassic), vicinity of Solothurn, Canton of Solothurn, Switzerland (Fig. 1).

Illustrations of type. Rütimeyer (1873: plate V, Fig. 1); Figs. 3A–3D.

Figure 3 Craspedochelys picteti.

Holotype of C. picteti, NMS 129: (A) photograph of the carapace; (B) interpretative drawing of the carapace; (C) photograph of the plastron; (D) interpretative drawing of the plastron. Referred specimen, NMS 608: (E) photograph of the carapace; (F) interpretative drawing of the carapace. Bones are white; green solid lines indicate scales sulci; matrix is gray. Abbreviations: ca, carapace; cpf, central plastral fontanelle; hyo, hyoplastron; p, peripheral; pla, plastron; py, pygal; sp, suprapygal; *, intermediate element (see text).

Referred specimens. Specimens listed in Bräm (1965); NMS 130 (holotype of Craspedochelys crassa Rütimeyer, 1873.

Revised diagnosis. Craspedochelys picteti can be diagnosed as a representative of Craspedochelys by a broad carapace, about as wide as long (as preserved), a weak nuchal notch, and a high length/width ratio of costal bones. Differing from C. jaccardi in: greater size (carapace length up to 550 mm); carapace heraldic shield-shaped and more quadrangular anteriorly; slightly lower length/width ratio of costal bones (4.3 as opposed to 4.8 or more for the fourth costal in C. jaccardi); relatively small pygal; contact between peripheral 11 and costal 8 limited or absent; hyoplastron slightly longer than wide.

Remarks. Bräm (1965) notably characterized C. picteti by the following suite of features: carapace as wide as long and shaped like a heraldic shield, with anterior rim extending only slightly convex up to the third peripheral then bending almost at right angle toward the rear; nuchal notch weak; vertebral scales moderately broad extending only about one third of the length of costals; free first thoracic rib; second thoracic rib contacting only the second thoracic vertebra. Gaffney (1975a) tentatively synonymized C. picteti with P. etalloni, considering the features proposed by Bräm (1965) as resulting either from postmortem deformation (carapace shape and width) or from biological variation (degree of nuchal emargination, width of vertebral scales). Furthermore, Gaffney (1975a) argued that the first and second thoracic ribs are only visible in NMS 608, and that their condition is ambiguous due to incomplete preparation and postmortem damage. NMS 608 is currently mounted on a wall in the NMS exhibition, and we were therefore unable to confirm Bräm’s (1965) observations. Antunes, Becquart & Broin (1988) rejected Gaffney’s (1975a) conclusions and synonymized C. picteti with C. jaccardi, though without directly studying the Swiss material. Finally, Lapparent de Broin, Lange-Badré & Dutrieux (1996) re-established C. jaccardi and C. picteti as distinct species, considering the first as a smaller form with thinner shell plates.

NMS 129, NMS 608, and NMS 130 (holotype of C. crassa) share a number of features that clearly distinguish them from other species from the Jura Mountains: anterior part of the carapace broad with anterior rim almost straight up to the level of the p3–p4 suture; reduced nuchal notch restricted to the nuchal plate; second and third vertebral scales extending about one third of the length of the costals (Fig. 3). These are the same characters (Gaffney, 1975a) dismissed as resulting from postmortem deformation or biological variation. However, our review of the Solothurn material indicates that these features are never found in any other specimen, no matter how deformed or variable it may be (see Discussion). Therefore, we consider Bräm’s (1965) conclusions on C. picteti as valid. However, this species is relatively poorly known and more material is needed.

Craspedochelys crassa Rütimeyer, 1873

1873 Craspedochelys crassa. Rütimeyer [new species]

1965 Craspedochelys picteti. Bräm [subjective synonymy]

Taxonomic assessment. Invalid name, subjective synonym of Craspedochelys picteti Rütimeyer, 1873.

Type material. NMS 130, a poorly preserved carapace fragment. Holotype by monotypy Bräm (1965: 139).

Type horizon and locality. Solothurn Turtle Limestone, uppermost member of the Reuchenette Formation (Kimmeridgian, Late Jurassic), vicinity of Solothurn, Canton of Solothurn, Switzerland (Fig. 1).

Illustrations of type. Rütimeyer (1873: plate IX, Figs. 5 and 5b).

Remarks. The illustration published by Rütimeyer (1873) greatly improves on the actual specimen (NMS 130), whose state of preservation is rather poor. However, the anterior outline of the carapace and the vertebral covering less than half of the length of the costals correspond to what is known in C. picteti. Rütimeyer (1873) distinguished C. picteti and C. crassa based on the greater thickness of the costal bones in the latter, which is, as pointed out by Bräm (1965: 137), a feature subject to a certain level of individual variations. We see no reason to separate the two species based on the available material and agree with Bräm (1965) in synonymizing C. crassa with C. picteti.

Craspedochelys plana Rütimeyer, 1873

1873 Craspedochelys plana. Rütimeyer [new species]

1965 Tropidemys langii. Bräm [subjective synonymy]

Taxonomic assessment. Invalid name, subjective synonym of Tropidemys langii Rütimeyer, 1873.

Type material. NMS 132, anterolateral (left) portion of a carapace. Holotype by monotypy (Bräm, 1965: 183).

Type horizon and locality. Solothurn Turtle Limestone, uppermost member of the Reuchenette Formation (Kimmeridgian, Late Jurassic), vicinity of Solothurn, Canton of Solothurn, Switzerland (Fig. 1).

Illustrations of type. Rütimeyer (1873: plate IX, Figs. 1 and 2).

Remarks. Bräm (1965: 184) concluded that NMS 132 should in fact be assigned to Tropidemys langii. What is preserved of the dorsal surface of the carapace does not allow a definitive attribution to either C. picteti or Tr. langii. However, the visceral surface of costal 1 clearly shows a crest-like axillary buttress, a feature characteristic of Tr. langii (Püntener et al., 2014). We therefore follow Bräm (1965) and Püntener et al. (2014) in referring this specimen to Tr. langii.

Craspedochelys jaccardi (Pictet, 1860)

1860 Emys jaccardi. Pictet [new species]

1873 Plesiochelys Jaccardi. Rütimeyer [new combination]

1975a Plesiochelys etalloni. Gaffney [subjective synonymy]

1988 Craspedochelys jaccardi. Antunes, Becquart & Broin [new combination]

Taxonomic assessment. Valid name.

Synonymy. None.

Type material. MHNN FOS 977, a complete shell. Holotype by monotypy.

Type horizon and locality. Les Hauts-Geneveys, Canton of Neuchâtel, Switzerland (Fig. 1), “Virgulien supérieur”, possibly corresponding to the early Tithonian (see Lapparent de Broin, Lange-Badré & Dutrieux, 1996: 552). According to Pictet (1860), the specimen was collected from a quarry near Les Brenets, whereas for Jaccard (1860) the specimen came from a different quarry near Les Hauts-Geneveys. Jaccard (1870) confirmed the locality as being Les Hauts-Geneveys (Ayer, 1997).

Illustrations of type. Pictet (1860: plates I–III); Lapparent de Broin, Lange-Badré & Dutrieux (1996: plate IV); Figs. 4A–4D.

Figure 4 Craspedochelys jaccardi.

Holotype of C. jaccardi, MHNN FOS 977: (A) photograph of the carapace; (B) interpretative drawing of the carapace; (C) photograph of the plastron; (D) interpretative drawing of the plastron. Referred specimen, NMS 101: (E) photograph of the carapace; (F) interpretative drawing of the carapace; (G) photograph of the plastron; (H) interpretative drawing of the plastron. Referred specimen, NMS 673: (I) photograph of the carapace; (J) interpretative drawing of the carapace; (K) photograph of the plastron; (L) interpretative drawing of the plastron. Referred specimen, NMS 102a: (M) photograph of the carapace; (N) interpretative drawing of the carapace. Bones are white; green solid lines indicate scales sulci; matrix is gray. Abbreviations: co, costal; n, neural; p, peripheral; *, intermediate element (see text).

Referred specimens. See Bräm (1965).

Revised diagnosis. Craspedochelys jaccardi can be diagnosed as a representative of Craspedochelys by a broad carapace, about as wide as long (as preserved), a weak nuchal notch, a high length/width ratio of costal bones, and a hyoplastron wider than long. Differing from C. picteti in: smaller size (carapace length up to 420 mm); carapace more evenly rounded anteriorly; higher length/width ratio of costal bones (4.8 or more, as opposed to 4.3 for the fourth costal); wider pygal bone; contact between peripheral 11 and costal 8 present; hyoplastron wider than long (slightly longer than wide in C. picteti).

Remarks. Emys jaccardi Pictet, 1860 was referred to the genus Plesiochelys by Rütimeyer (1873), a conclusion followed by Bräm (1965), who differentiated this species based mainly on the following features: carapace about as wide as long; nuchal notch evenly rounded; plastron oval in outline; small xiphiplastral notch; plastron length about 73% that of the carapace; small central plastral fontanelle, mainly formed by hypoplastron; vertebrals relatively narrow. Gaffney (1975a) synonymized P. jaccardi with P. etalloni, notably explaining the broad shell of the former by postmortem compression. Antunes, Becquart & Broin (1988) were the first to refer the species jaccardi to the genus Craspedochelys based on the following characters: broad carapace (width/length ratio exceeding 90%); pentagonal outline with anterior part quadrangular; small central plastral fontanelle; and vertebral scales reduced in width. This position was later confirmed by Lapparent de Broin, Lange-Badré & Dutrieux (1996), although, in contrast to Antunes, Becquart & Broin (1988), they recognized C. picteti and C. jaccardi as two distinct forms, based primarily on size difference and variation in the thickness of the shell bones.

The characteristics exhibited by the holotype of C. jaccardi (MHNN FOS 977) are inconsistent with a referral to P. etalloni, as suggested by Gaffney (1975a). For example, postmortem compression or individual variation cannot explain the significant reduction of the plastron length in C. jaccardi (Table 2). The proportions of costals, hyoplastron, and xiphiplastron are also markedly different in the two species (see Discussion). Therefore, C. jaccardi is considered as a valid species. However, this species is only known by a limited number of specimens and some questions remain regarding the attribution of the Solothurn specimens to this species (see Discussion).

Table 2 Comparison of the ratio between the length of the plastron and the length of the carapace in selected specimens referred to P. etalloni, C. picteti, and C. jaccardi.

	Plastron
length (mm)	Carapace
length (mm)	Ratio	
Plesiochelys etalloni				
NMS 59	400	474	0.84	
NMS 78	–	361a	–	
NMS 79	–	–	–	
NMS 116	–	–	–	
NMS 669	363	410	0.89	
NMS 675	369	445	0.83	
MAJ 2005-11-1	431	471	0.92	
Craspedochelys jaccardi				
NMS 101	300b	413	0.73	
NMS 102a	–	363	–	
NMS 612	–	–	–	
NMS 673	292	411	0.71	
MHNN FOS 977	283	365	0.78	
Craspedochelys picteti				
NMS 608	–	540	–	
Notes.

a Carapace missing about 20 mm.

b Estimated plastron length.

Tropidemys Rütimeyer, 1873

1873 Tropidemys. Rütimeyer [new genus]

Type species. Tropidemys langii Rütimeyer, 1873.

Revised diagnosis. See Püntener et al. (2014)

Remarks. Tropidemys is mainly characterized by wide, hexagonal and often keeled neurals. The validity of this genus has never been questioned. A recent review is available in Püntener et al. (2014).

Tropidemys langii Rütimeyer, 1873

1873 Tropidemys Langii. Rütimeyer [new species]

Taxonomic assessment. Valid name.

Synonymy. Tropidemys expansa Rütimeyer, 1873, Tropidemys gibba Rütimeyer, 1873, and Craspedochelys plana Rütimeyer, 1873.

Type material. NMS 16, posterior part of a carapace. Lectotype designated by Bräm (1965: 176).

Type horizon and locality. Solothurn Turtle Limestone, uppermost member of the Reuchenette Formation (Kimmeridgian, Late Jurassic), vicinity of Solothurn, Canton of Solothurn, Switzerland (Fig. 1).

Illustrations of type. Rütimeyer (1873: plate VII, Fig. 1); Figs. 5A and 5B.

Figure 5 Tropidemys langii.

Lectotype of Tr. langii, NMS 16: (A) photograph of the carapace; (B) interpretative drawing of the carapace. Referred specimen, NMS 15: (C) photograph of the carapace; (D) interpretative drawing of the carapace. Bones are white; green solid lines indicate scales sulci; matrix is gray. Abbreviations: co, costal; n, neural; sp, suprapygal; *, intermediate element (see text).

Referred specimens. See Püntener et al. (2014).

Revised diagnosis. See Püntener et al. (2014).

Remarks. Rütimeyer (1873) initially described three species of Tropidemys in Solothurn: Tr. langii, Tr. expansa and Tr. gibba. Bräm (1965), who had access to a sub-complete carapace (NMS 15; Figs. 5C and 5D), concluded that there was no reason to differentiate three species based on the available material. Püntener et al. (2014) recently revised the Solothurn material and described new specimens from the Kimmeridgian in the vicinity of Porrentruy, Switzerland (Fig. 1). They confirmed Bräm’s (1965) conclusions.

Tropidemys expansa Rütimeyer, 1873

1873 Tropidemys expansa. Rütimeyer [new species]

1965 Tropidemys langii. Bräm [subjective synonymy]

2014 Tropidemys langii. Püntener et al. [subjective synonymy]

Taxonomic assessment. Invalid name, subjective synonym of Tropidemys langii Rütimeyer, 1873.

Type material. Rütimeyer (1873) did not explicitly refer to a type specimen in his description of Tr. expansa. However, he figured specimens NMS 32 and NMS 33 (Rütimeyer, 1873: plate IX, Figs. 3–4), and they, at least, form part of the syntype series.

Type horizon and locality. Solothurn Turtle Limestone, uppermost member of the Reuchenette Formation (Kimmeridgian, Late Jurassic), vicinity of Solothurn, Canton of Solothurn, Switzerland (Fig. 1).

Illustrations of type. Rütimeyer (1873: plate IX, Figs. 3–4).

Remarks. Püntener et al. (2014) revised this material and concluded that Tr. expansa was a junior subjective synonym of Tr. langii.

Tropidemys gibba Rütimeyer, 1873

1873 Tropidemys gibba. Rütimeyer [new species]

1965 Tropidemys langii. Bräm [subjective synonymy]

2014 Tropidemys langii. Püntener et al. [subjective synonymy]

Taxonomic assessment. Invalid name, subjective synonym of Tropidemys langii Rütimeyer, 1873.

Type material. NMS 38, a fragment of carapace with neurals 3–6 and medial parts of associated costals. Holotype designated by Rütimeyer (1873).

Type horizon and locality. Solothurn Turtle Limestone, uppermost member of the Reuchenette Formation (Kimmeridgian, Late Jurassic), vicinity of Solothurn, Canton of Solothurn, Switzerland (Fig. 1).

Illustrations of type. Rütimeyer (1873: plate IV, Fig. 1) and Bräm (1965: plate VIII, Fig. 5).

Remarks. Püntener et al. (2014) studied this material and concluded that Tr. gibba was a junior subjective synonym of Tr. langii.

THALASSEMYDIDAE Zittel, 1889

Thalassemys Rütimeyer, 1859a

1859a Thalassemys. Rütimeyer [new genus].

Type species. Thalassemys hugii Rütimeyer, 1873.

Revised diagnosis. Differing from Plesiochelys, Craspedochelys, and Tropidemys in: great anterior widening of first neural; presence of small costo-peripheral fontanelles in the adults; presence of clearly visible linear striations perpendicular to sutures between most shell elements; presence of a lateral plastral fontanelle; non-sutural connection of the epiplastron and entoplastron with the hyoplastron; presence of a small xiphiplastral fontanelle.

Remarks. Bräm (1965) diagnosed Thalassemys mainly based on the following combination of features: carapace relatively flat and more or less heart-shaped in outline; shell moderately high, the height being mostly the result of the ascending processes of the hyo- and hypoplastra; costo-peripheral fontanelles retained in adult individuals; one cervical scale; large central plastral fontanelle (extending over most of the length of the plastron in Th. hugii); lateral plastral fontanelle absent. According to Bräm (1965), Thalassemys includes Th. hugii and Th. moseri, but not Th. marina Fraas, 1903 from the Tithonian of Schnaitheim, Germany (a form he referred to Eurysternum on the account of the presence of a lateral plastral fontanelle).

Our review of the Solothurn material indubitably establishes that a lateral plastral fontanelle was indeed present in Th. hugii, the type species of Thalassemys. Additionally, we were able to reassess the plastral morphology of Th. hugii (see Discussion). Based notably on these new data, ‘Th.’ moseri is excluded from Thalassemys (see below), whereas Th. marina is consistent with our concept of Thalassemys.

Thalassemys hugii Rütimeyer, 1873

1873 Thalassemys Hugii. Rütimeyer [new species]

Taxonomic assessment. Valid name.

Synonymy. Thalassemys Gresslyi Rütimeyer, 1873 and Eurysternum ignoratum Bräm, 1965.

Type material. NMS 1, a large carapace plus associated plastron fragments and postcranial remains. Lectotype designated by Bräm (1965: 143).

Type horizon and locality. Solothurn Turtle Limestone, uppermost member of the Reuchenette Formation (Kimmeridgian, Late Jurassic), vicinity of Solothurn, Canton of Solothurn, Switzerland (Fig. 1).

Illustrations of type. Rütimeyer (1873: plate I); Bräm (1965: plate 7); Figs. 6A–6D.

Figure 6 Thalassemys hugii.

Lectotype of Th. hugii, NMS 1: (A) photograph of the carapace; (B) interpretative drawing of the carapace; (C) photograph of the plastron; (D) interpretative drawing of the plastron. Holotype of Eurysternum ignoratum, NMS 5: (E) photograph of costals 4–6; (F) interpretative drawing of costals 4–6; (G) photograph of the hyoplastra; (H) interpretative drawing of the hyoplastra. Holotype of Th. gresslyi, NMS 12: (I) photograph of the carapace; (J) interpretative drawing of the carapace. Referred specimen, NMS 124: (K) photograph of the carapace; (L) interpretative drawing of the carapace. Referred specimen, NMS 412: (M) photograph of the carapace; (N) interpretative drawing of the carapace. Bones are white; green solid lines indicate scales sulci; matrix is gray. Abbreviations: co, costal; hyo, hyoplastron; hypo, hypoplastron; n, neural; sp, suprapygal; v, vertebral scale; xi, xiphiplastron; *, intermediate element (see text).

Referred specimens. Specimens listed in Bräm (1965); NMS 5 (holotype of Eurysternum ignoratum Bräm, 1965), NMS 12 (holotype of Th. gresslyi Rütimeyer, 1873), NMS 124, NMS 412, NMS 20981, NMS 22325-22327 (and associated remains), NMS 37251.

Revised diagnosis. Thalassemys hugii can be diagnosed as a representative of Thalassemys by the great widening of neural 1, the retention of costo-peripheral fontanelles, the presence of clearly visible linear striations perpendicular to sutures between most shell elements, the presence of a lateral plastral fontanelle, the absence of sutural connection of the epi- and entoplastron with the hyoplastron, and the presence of a small xiphiplastral fontanelle. Differing from Th. marina in: narrower vertebral scales with anterolateral and posterolateral margins of equal length (as opposed to posterolateral margin shorter in Th. marina); smaller lateral plastral fontanelle.

Remarks. Because Thalassemys hugii was typified based on a relatively completed shell and partial associated post-cranial remains, its validity has never been questioned. This turtle is nonetheless not very well known and remains are relatively rare in contemporaneous deposits. This is the largest turtle in Solothurn reaching more than 630 mm (the pygal is missing in NMS 1). Bräm (1965) diagnosed Th. hugii by the following features: presence of a large longitudinal central plastral fontanelle, sometimes closed anteriorly by the hyoplastra and posteriorly by the xiphiplastra; vertebral scales relatively narrow (as opposed to wide in ‘Th.’ moseri); very large size.

During our review of the Solothurn material, we were able to identify a set of characters that prompted a revision of the traditional concept of Th. hugii (see Discussion). Perhaps the most important of these characters is the presence of a lateral plastral fontanelle, the purported absence of which was used by Bräm (1965) to differentiate Thalassemys from Eurysternum. Based on this review of the material, E. ignoratum is synonymized with Th. hugii (see below).

Thalassemys gresslyi Rütimeyer, 1873

1873 Thalassemys Gresslyi. Rütimeyer [new species]

1965 Thalassemys hugii. Bräm [subjective synonymy]

Taxonomic assessment. Invalid name, subjective synonym of Thalassemys hugii Rütimeyer, 1873.

Type material. NMS 12, anterior half of a large carapace partly disarticulated. Holotype by monotypy (see Bräm, 1965: 152).

Type horizon and locality. Solothurn Turtle Limestone, uppermost member of the Reuchenette Formation (Kimmeridgian, Late Jurassic), vicinity of Solothurn, Canton of Solothurn, Switzerland (Fig. 1).

Illustrations of type. Figs. 6I–6J.

Remarks. Rütimeyer (1873) argued that NMS 12 should be assigned to a separate species because of differences in proportions of the first neural and first costal, a larger size and a different sculpturing of the bone surface compared to Th. hugii. However, Bräm (1965: 152) concluded that these differences could be explained either by individual variations or postmortem deformation and synonymized Th. gresslyi with Th. hugii. The state of preservation of this specimen is rather poor and the sculpturing of the bone surface is undoubtedly of postmortem origin. The size of the specimen and the shape of the vertebral scales are consistent with our concept of Th. hugii. Therefore, we follow Bräm’s (1965) conclusion.

‘Eurysternum’ ignoratum Bräm, 1965

1965 Eurysternum ignoratum. Bräm [new species]

Taxonomic assessment. Invalid name, subjective synonym of Thalassemys hugii Rütimeyer, 1873.

Type material. NMS 5, disarticulated and fragmentary remains (three costals, hyoplastra, scapulae, humerus, pubes). Holotype (Bräm, 1965: 166).

Type horizon and locality. Solothurn Turtle Limestone, uppermost member of the Reuchenette Formation (Kimmeridgian, Late Jurassic), vicinity of Solothurn, Canton of Solothurn, Switzerland (Fig. 1).

Illustrations of type. Rütimeyer (1873: plate VI, Fig. 4); Bräm (1965: plate 8, Fig. 6); Figs. 6E–6H.

Previously referred specimens. NMS 124 and NMS 412 (see Bräm, 1965). The NMS catalogue also assigns four additional specimens to E. ignoratum. NMS 20981 and NMS 37251 are herein referred to Th. hugii. NMS 21908 and NMS 21922 consist of isolated bones that have been included in resin and sampled for histological analysis prior to the present study, which prevents proper examination.

Remarks. Bräm (1965) identified Eurysternum ignoratum as a representative of Eurysternum based on the presence of a lateral plastral fontanelle. He differentiated E. ignoratum from E. wagleri by the presence of narrow vertebral scales. Broin (1994) and Lapparent de Broin, Lange-Badré & Dutrieux (1996) tentatively suggested a possible synonymy between E. ignoratum and Solnhofia parsonsi Gaffney, 1975b, but the material referred to E. ignoratum was never actually revised in detail since Bräm (1965).

Our review of the concerned material leads us to the conclusion that there is not a single character that differentiates E. ignoratum from Th. hugii (see Discussion). Eurysternum ignoratum is therefore interpreted as a subjective junior synonym of Th. hugii.

‘Thalassemys’ moseri (Bräm, 1965)

1965 Thalassemys moseri. Bräm [new species]

1996 Plesiochelys solodurensis. Lapparent de Broin, Lange-Badré & Dutrieux [subjective synonymy]

Taxonomic assessment. Valid name.

Synonymy. None.

Type material. NMS 618, partial carapace and plastron. Holotype (Bräm, 1965: 155).

Type horizon and locality. Solothurn Turtle Limestone, uppermost member of the Reuchenette Formation (Kimmeridgian, Late Jurassic), vicinity of Solothurn, Canton of Solothurn, Switzerland (Fig. 1).

Illustrations of type. Bräm (1965: plate 8, Figs. 2 and 3); Figs. 7A–7D.

Figure 7 ‘Thalassemys’ moseri.

Holotype of ‘Th.’ moseri, NMS 618: (A) photograph of the carapace; (B) interpretative drawing of the carapace; (C) photograph of the plastron; (D) interpretative drawing of the plastron. Referred specimen, NMS 62: (E) photograph of the carapace; (F) interpretative drawing of the carapace. Bones are white; stripped lines indicate internal bone layers; green solid lines indicate scales sulci; matrix is gray. Abbreviations: ce, cervical scale; co, costal; hyo, hyoplastron; n, neural; nu, nuchal.

Referred specimens. Specimens listed in Bräm (1965); PMZH A/III 514, a nearly complete skull and partial shell from the early Tithonian of La Morelière (Isle of Oléron, Department of Charente-Maritime, France) referred by Rieppel (1980).

Revised diagnosis. Species of dubious affinity characterized by the combination of the following features: medium-sized shell (carapace length about 350 mm); three cervical scales; wide vertebrals covering about half of the costals laterally; pattern of carapacial scales recalling that of Plesiochelys and Craspedochelys; presence of costo-peripheral fontanelles; costal bones relatively thin distally; rib tips easily disarticulated from peripherals; large central plastral fontanelle, oval in outline; absence of lateral plastral fontanelle; epi- and entoplastron not sutured to hyoplastron; xiphiplastron possibly forming a small xiphiplastral notch. See Rieppel (1980) for cranial characters. Differing from Plesiochelys etalloni, Craspedochelys picteti, C. jaccardi, and Tropidemys langii in: retention of costo-peripheral fontanelles; epi- and entoplastron not sutured to hyoplastron. Differing from Thalassemys hugii in: smaller size; wide vertebrals covering about half of the costals laterally; absence of lateral plastral fontanelle.

Remarks. Bräm (1965) originally diagnosed ‘Thalassemys’ moseri as a representative of Thalassemys by the retention of costo-peripheral fontanelles and the absence of lateral plastral fontanelles. ‘Thalassemys’ moseri was furthermore differentiated from Th. hugii by its smaller size, the presence of broad vertebral scales, and the presence of a large central plastral fontanelle closed anteriorly by the hyoplastra and posteriorly by the hypoplastra (as opposed to the very extensive central fontanelle he considered to be present in Th. hugii; but see Discussion). Lapparent de Broin, Lange-Badré & Dutrieux (1996) argued that the holotype of ‘Th.’ moseri (NMS 618) with its three cervicals, wide vertebrals and oval central plastral fontanelle was probably a young individual of P. solodurensis, the only Plesiochelys species they recognized in Solothurn. According to them, this specimen could not be referred to Thalassemys because the type species of this genus is a very large form with only one cervical scale. Bräm (1965: 155) himself was aware of the apparent similarities between ‘Th.’ moseri and what he described as P. etalloni (i.e., Plesiochelys specimens with a central plastral fontanelle). This is especially true for the pattern of carapacial scales. However, ‘Th.’ moseri and P. etalloni diverge on characters that are traditionally used to differentiate plesiochelyids from thalassemydids, such as the presence of carapacial fontanelles between costals and peripherals.

Our review of the material clearly indicates that ‘Th.’ moseri is unique among Solothurn turtles. The presence of costo-peripheral fontanelles and the absence of sutural contact between the hyoplastron and the epiplastron and entoplastron are inconsistent with an attribution to P. etalloni, C. picteti, or C. jaccardi. The elongated neurals, the absence of a neural keel, and the broad vertebral scales clearly differentiate ‘Th.’ moseri from Tr. langii. Finally, the smaller size, the broad vertebrals, and the absence of a lateral plastral fontanelle distinguish ‘Th.’ moseri from Th. hugii. Therefore, we confirm the conclusions of Bräm (1965) and consider ‘Th.’ moseri as a distinct species. However, the generic attribution to Thalassemys is rejected. The pattern of carapacial scales and the absence of lateral plastral fontanelle suggest that this species is more closely related to Plesiochelys than to Thalassemys. However, the skull described by Rieppel (1980) is clearly different from that of P. etalloni, which prevents a tentative referral of ‘Th.’ moseri to Plesiochelys (see Discussion). The material described by Rieppel (1980) should be duly revised.

EURYSTERNIDAE Dollo, 1886

Type genus

Eurysternum Meyer, 1839.

Remarks. A discussion on the genus Eurysternum and a reevaluation of its type species E. wagleri are available in Anquetin & Joyce (in press). Since Eurysternum ignoratum is herein interpreted as a junior synonym of Thalassemys hugii (see above), the fossil record of eurysternids in the Late Jurassic of the Jura Mountains is now limited to a single skull from Solothurn referred to Solnhofia parsonsi (Gaffney, 1975b).

DISCUSSION

Plesiochelys etalloni

Plesiochelys etalloni is known from about 30 relatively complete shells and uncountable shell fragments, most of which from the quarries in the vicinity of Solothurn, Switzerland (Fig. 1). This material provides a good opportunity to grasp the level of intraspecific variability in this fossil species. A general description of the shell morphology of P. etalloni can be found in Anquetin, Deschamps & Claude (2014).

The carapace of P. etalloni is usually evenly oval, but some specimens have a more quadrangular anterior rim (e.g., NMS 78 and NMS 116; Fig. 8). Carapaces that have been flattened during fossilization tend to be characterized by a more pronounced angulation of their anterior margin, resulting from the partial disarticulation of some peripherals. In P. etalloni and Craspedochelys jaccardi (MHNN FOS 977; Fig. 4), this angulation is always located at the level of the p2–p3 suture, whereas C. picteti (NMS 129 and NMS 608; Fig. 3) is unique in showing an angulation at the level of the p3–p4 suture (see below).

Figure 8 Intraspecific variability in Plesiochelys etalloni.

NMS 78: (A) photograph of the carapace; (B) interpretative drawing of the carapace; (C) photograph of the plastron; (D) interpretative drawing of the plastron. NMS 79: (E) photograph of the carapace; (F) interpretative drawing of the carapace; (G) photograph of the plastron; (H) interpretative drawing of the plastron. NMS 669: (I) photograph of the carapace; (J) interpretative drawing of the carapace; (K) photograph of the plastron; (L) interpretative drawing of the plastron. NMS 675: (M) photograph of the carapace; (N) interpretative drawing of the carapace; (O) photograph of the plastron; (P) interpretative drawing of the plastron. NMS 116: (Q) photograph of the carapace; (R) interpretative drawing of the carapace; (S) photograph of the plastron; (T) interpretative drawing of the plastron. NMS 94: (U) photograph of the plastron; (V) interpretative drawing of the plastron. NMS 629: (W) photograph of the plastron; (X) interpretative drawing of the plastron. Bones are white; stripped lines indicate internal bone layers; green solid lines indicate scales sulci; support material is brown; matrix is gray. Abbreviations: eb, epiplastral bulb; n, neural; sp, suprapygal; *, intermediate element (see text).

The posteromedial region of the carapace is relatively variable in P. etalloni, as generally common in turtles (Zangerl, 1969). The seventh and eighth neurals are usually shorter and more variable in shape than the preceding ones. These two neurals even fuse in some specimens (e.g., NMS 79 and NMS 669; Fig. 8). The eighth neural might even be much reduced or absent in certain individuals allowing a midline contact of the eighth costals (e.g., MAJ 2005-11-1; Fig. 2; see Anquetin, Deschamps & Claude, 2014). In most specimens, there is an intermediate element of varying size and shape between the eighth neural and the first suprapygal (Figs. 2 and 8). We are uncertain of the identity of this additional element (ninth neural, additional suprapygal, or neomorphic bone). Its shape and size are quite variable, from a small quadrangular element about the size of preceding neurals to a large triangular or trapezoidal element about the size of the following suprapygal. This extreme variation of size and shape is probably inconsistent with an identification as a ninth neural, but this intermediate element is also articulated with the vertebral series (at least partially), which is incongruent with an identification as a suprapygal. For the time being, we prefer to simply refer to this element as the ‘intermediate’ element. It is particularly interesting to note that a similar element in known in C. picteti (Fig. 3), C. jaccardi (Fig. 4), Tropidemys langii (Fig. 5), and Thalassemys hugii (Fig. 6). The fourth intervertebral sulcus always runs on this intermediate element, or on the first suprapygal if the intermediate element is absent. Posterior to the intermediate element, there are usually two suprapygals, which sometimes fuse into one single element. The first suprapygal is generally larger and wider than the second, preventing a contact between the latter and the eighth pair of costals, but the actual size of each suprapygal is relatively variable from one individual to another (Fig. 8).

The three cervical scales are visible in all specimens in which this area is sufficiently preserved, but it should be noted that the cervical sulci are lost relatively quickly once the area is slightly damaged. The presence of three cervicals has long been though to represent a unifying character of Plesiochelyidae sensu stricto, including Plesiochelys, Craspedochelys, and Tropidemys (e.g., Bräm, 1965; Lapparent de Broin, Lange-Badré & Dutrieux, 1996; Slater et al., 2011; Püntener et al., 2014; Pérez-García, in press). However, the presence of three cervicals has been reported in some Eurysternidae (Joyce, 2003; Anquetin & Joyce, in press), and three cervicals may also have been present in Th. hugii (see below).

The pattern of carapacial scales of Plesiochelys etalloni is similar to that of C. picteti, C. jaccardi, and ‘Th.’ moseri. In P. etalloni, this pattern is subject to a certain degree of variability. This informs us on the variability that may be expected in the other aforementioned species, which are currently represented by considerably less specimens. The vertebral sulci are generally sinuous, but to a variable extent from one individual to another. Vertebrals 2–4 are wide, hexagonal scales. However, if the second and third vertebrals consistently cover about half of the costal length laterally, the lateral extent of the fourth vertebral is more variable. In some specimens (e.g., NMS 79 and NMS 118; Figs. 2 and 8), the fourth vertebral extends as far as the peripherals laterally, significantly reducing the width of the fourth pleural in the process. The marginals are generally restricted to the peripherals, but in some specimens the fourth and/or seventh marginals extend very slightly on the costals (e.g., NMS 59, NMS 60, and NMS 669; Figs. 2 and 8). Finally, it is interesting to note that, in all specimens in which this area is known, the twelfth pair of marginals extends anteriorly on the second suprapygal, whereas it is restricted to the pygal in C. picteti and C. jaccardi (not known in ‘Th.’ moseri).

Bräm (1965) mentioned the presence of epiplastral bulbs in P. etalloni. We confirm that two pairs of epiplastral bulbs are present in specimens with undamaged epiplastra (e.g., NMS 59, NMS 94, NMS 629, NMS 669; Figs. 2 and 8). The entoplastron is usually diamond-shaped, but in some specimens its posterior half is more or less elongated. Finally, the presence/absence of a central plastral fontanelle is interpreted as an intraspecific variation of P. etalloni (Gaffney, 1975a; Lapparent de Broin, Lange-Badré & Dutrieux, 1996; Anquetin, Deschamps & Claude, 2014). Like their carapacial counterparts, the plastral scales exhibit a certain degree of variability in their shape and relations with underlying bony elements. In some specimens (e.g., MAJ 2005-11-1 and NMS 94; Figs. 2 and 8), the plastral midline sulcus is irregularly sinuous. The length of the pectoral compared to that of the humeral is quite variable in P. etalloni. The pectoral may be shorter (e.g., MAJ 2005-11-1, NMS 59, NMS 94), about equal (e.g., NMS 66, NMS 669), or longer than the humeral (e.g., NMS 629, NMS 675). Most commonly, there are four pairs of inframarginals, except in NMS 78 where there are five pairs (Fig. 8). These inframarginals are either entirely restricted to plastral elements (e.g., NMS 59, NMS 79), or some of them, usually the third and/or fourth, may extend slightly laterally on the peripherals (e.g., MAJ 2005-11-1, NMS 94). Finally, the anal scales very rarely extend anteriorly on the hypoplastra (e.g., NMS 59, NMS 79), otherwise the anals are restricted to the xiphiplastra (Figs. 2 and 8).

Craspedochelys picteti

Craspedochelys picteti is known mainly from two specimens from Solothurn (Fig. 1). The holotype (NMS 129) is relatively incomplete, consisting only of the anterior left quarter of the carapace and associated hyoplastra, but NMS 608 consists of a large, sub-complete carapace (Fig. 3). Craspedochelys picteti is mainly characterized by a heraldic shield-shaped carapace. Anteriorly, the carapace rim is almost straight transversally from the nuchal to the third peripheral. The carapace margin then bends abruptly posteriorly at the level of the p3–p4 suture. As discussed above for P. etalloni, variations in the degree of angulation of the anterior part of the carapace are probably the result of postmortem compression in these turtles, but the shift in the location of this angulation in C. picteti indicates that the anterior outline of the carapace was truly broader in this taxon. From peripherals 4 to 7, the margin is almost straight and parallel to the anteroposterior axis of the carapace. At the level of the p7–p8 suture, the margin bends abruptly medially and continues obliquely toward the pygal. The width of the carapace decreases rapidly from the eighth peripheral to the pygal.

The nuchal is a wide, trapezoidal element with a shallow nuchal notch, which does not extend on the first peripheral. Specimens referred to P. etalloni usually have a more pronounced and more laterally extended nuchal notch. There are eight neurals. The first neural is more rectangular. Neurals 2–6 are elongate, hexagonal elements with shorter sides facing anteriorly. As in P. etalloni, there was probably a certain amount of intraspecific variability in the morphology of the seventh and eighth neurals. In NMS 608, neural 7 is a short hexagonal element, whereas neural 8 is an irregularly shaped, wider than long element. Posterior to neural 8, there is a large trapezoidal element that corresponds to the intermediate element described in P. etalloni (see above). In NMS 608, the two suprapygals may have been fused together, but poor preservation prevents a definitive conclusion on the matter. The pygal is a relatively small, almost square-shaped element. In P. etalloni and C. jaccardi, the pygal is usually much wider (Figs. 2 and 4). Probably as a result of the reduced size of the pygal, the eleventh peripheral does not contact the eighth costal in NMS 608. It is uncertain whether this unique configuration of the pygal area is a true characteristic of C. picteti or an individual variation of NMS 608, but NMS 61 (an indeterminate carapace fragment) exhibits the exact same arrangement. There are eight pairs of costals. The length of costals 6–8 decreases rapidly posteriorly. Proportionally to their length the costals of C. picteti are thinner than those of P. etalloni, but not as much as those of C. jaccardi (Table 3). The arrangement and shape of carapacial scales remind that of P. etalloni, to the notable exception that vertebral scales are narrower and cover about a third to half of the costal length. However, this character appears to be subject to a significant amount of variation in P. etalloni. There are three cervical scales. The twelve pairs of marginals never extend on the costals. In contrast to P. etalloni, the twelfth marginals do not extend anteriorly on the second suprapygal.

Table 3 Comparison of the length/width ratio of the fourth costal in selected specimens referred to P. etalloni, C. picteti, and C. jaccardi.

	Costal 4
length (mm)	Costal 4
width (mm)	Ratio	
Plesiochelys etalloni				
NMS 59	176	56	3.14	
NMS 78	152	46	3.30	
NMS 79	163	45	3.62	
NMS 116	–	–	–	
NMS 669	160	47	3.40	
NMS 675	178	49	3.63	
MAJ 2005-11-1	–	–	–	
Craspedochelys jaccardi				
NMS 101	188	38	4.95	
NMS 102a	164	32	5.13	
NMS 612	155	32	4.84	
NMS 673	181	38	4.76	
MHNN FOS 977	–	–	–	
Craspedochelys picteti				
NMS 608	217	50	4.34	

Our knowledge of the plastron of C. picteti is limited to the hyoplastron of NMS 129 (Figs. 3C and 3D). This element is slightly longer than wide, which contrasts with the condition in C. jaccardi (see below). Based on the shape of its sutural contact with the hyoplastron, the entoplastron was probably a small element. A central plastral fontanelle was present in NMS 129. There are no further indications on the shape and size of the plastron in this species, which prevents comparison with other taxa from the Jura Mountains, notably P. etalloni and C. jaccardi.

Craspedochelys jaccardi

Craspedochelys jaccardi was originally described based on a single shell (MHNN FOS 977) from the vicinity of Neuchâtel, Switzerland (Fig. 1). Subsequently, additional specimens from Solothurn have been referred to this species (Rütimeyer, 1873; Bräm, 1965), although they are characterized by a slightly divergent morphology. Therefore, the following discussion is primarily based on the morphology of the holotype (Figs. 4A–4D). Craspedochelys jaccardi is a moderately sized turtle (carapace length up to 420 mm) characterized notably by a shortened plastron representing less than 80% of the carapace length (as opposed to 85–90% in Plesiochelys etalloni; see Table 2). As preserved, the shell is broad, even as wide as long in some specimens. Postmortem compression may affect our perception of shell width, but none of the many Solothurn specimens referred to P. etalloni has a shell as wide as long, no matter how flattened it is. In contrast to what Gaffney (1975a) suggested, the specimens referred to C. jaccardi are not more dorsoventrally flattened than specimens referred to P. etalloni. Anteriorly, the carapace is evenly rounded with only a weak nuchal notch mostly restricted to the nuchal bone. The carapace is slightly pentagonal in outline. The nuchal is a broad, trapezoidal element. The first neural is rectangular, whereas following neurals tend to be elongate and hexagonal with shorter sides anteriorly. There are up to eight neurals, but several specimens exhibit a reduction or loss of the seventh and/or eighth neurals allowing a midline contact of the seventh and/or eighth costals. As in P. etalloni and C. picteti notably, there is usually an intermediate element between the last neural and the first suprapygal (see above). As in other species, this element is relatively variable in shape and size. There are usually two suprapygals, the first larger than the second. The pygal is a wider than long element, larger than the same bone in C. picteti. There are eight pairs of costals, which are proportionally thinner and longer (higher length/width ratio; see Table 3) than those of P. etalloni and C. picteti. There are 11 pairs of longer than wide peripherals greatly increasing in width posteriorly. The posteriormost peripherals may have been slightly wider than long. The arrangement and shape of carapacial scales remind that of P. etalloni, but there seems to be a greater variability in the outline of vertebral scales in C. jaccardi (see below). There are three cervical scales. The twelve pairs of marginals never extend on the costals. In contrast to P. etalloni, the twelfth marginals do not extend anteriorly on the second suprapygal.

As noted above, the plastron of C. jaccardi is greatly reduced in length compared to that of P. etalloni. This reduction seems to result mainly from the shortening of the posterior lobe, which is apparent from the long post-xiphiplastral space (Lapparent de Broin, Lange-Badré & Dutrieux, 1996). The exact outline of the anterior plastral lobe is uncertain because the epiplastra are damaged in all known specimens. The posterior lobe is broad and rounded. There is a small central plastral fontanelle. The epi-hyoplastral suture is mostly transversal. Similarly to P. etalloni, the entoplastron is a diamond-shaped, longer than wide element with its anterior sides shorter than the posterior ones, but there appears to be a great variation in the size of this element between individuals (Fig. 4). The hyoplastron is remarkable in being wider than long (see Table 4), which probably reflects both the increased width of the shell and the reduced length of the plastron. Similarly, the xiphiplastron is as wide as long, which contrast with the longer than wide element found in most other turtles. There is a weak xiphiplastral notch, barely visible in some specimens. The extragular scales are restricted to the epiplastra. It is uncertain whether or not the gulars extended onto the anteromedial part of the entoplastron. The pectoral is reduced in length compared to the humeral. The anal scales are restricted to the xiphiplastra. There are four inframarginals increasing in length posteriorly. All inframarginals but the first extend slightly over the peripheral laterally (only visible in NMS 673).

Table 4 Comparison of the length/with ratio of the hyoplastron in selected specimens referred to P. etalloni, C. picteti, and C. jaccardi.

	Hyoplastron
length (mm)	Hyoplastron
width (mm)	Ratio	
Plesiochelys etalloni				
NMS 59	168	145	1.16	
NMS 78	154	133	1.16	
NMS 79	152a	138	1.10	
NMS 116	166	147	1.13	
NMS 669	156b	130b	1.20	
NMS 675	168b	149	1.13	
MAJ 2005-11-1	183c	156c	1.17	
Craspedochelys jaccardi				
NMS 101	117	158d	0.74	
NMS 102a	–	–	–	
NMS 612	–	–	–	
NMS 673	122	156	0.78	
MHNN FOS 977	118	123	0.96	
Craspedochelys picteti				
NMS 608	–	–	–	
Notes.

a Length incomplete.

b Incorrect measurement in Bräm (1965).

c From 3D surface mesh (see Anquetin, Deschamps & Claude, 2014).

d Width incomplete.

As mentioned above, there is a certain number of differences between the holotype of C. jaccardi (MHNN FOS 977) and specimens from Solothurn referred to this species (notably NMS 101 and NMS 673). On the carapace, the most obvious differences concern the vertebral scales. The vertebral pattern of MHNN FOS 977 is somewhat unusual (Figs. 4A and 4B). The first vertebral is narrower than the nuchal bone posteriorly and it widens greatly anteriorly to reach the sulcus between the first and second marginal. The second vertebral is similarly narrow anteriorly and its anterolateral margin curves greatly toward the midline. The second and third intervertebral sulci are displaced anteriorly lying just anterior to the middle of neural 3 and neural 5, respectively (instead of just posterior to the middle of neural 3 and over the posterior part of neural 5 in most other turtles). Consequently, the third vertebral is shorter, whereas the fourth vertebral is significantly longer (Fig. 4). An unusually long fourth vertebral is also known in a referred specimen from the Kimmeridgian of Murat (Department of Lot, France; Lapparent de Broin, Lange-Badré & Dutrieux, 1996: Figs. 3 and 4). As a result of this unusual arrangement of the vertebrals, the second pleural of MHNN FOS 977 is shortened, whereas the third pleural is greatly lengthened. The vertebral pattern of NMS 101 is also relatively unusual (Figs. 4E–4F). Vertebral sulci are irregularly sinuous. The outline of vertebrals 2–4 is particularly odd, with notably a narrower, sub-quadrangular third vertebral. The fifth vertebral is significantly shorter than in other specimens referred to C. jaccardi. The vertebral pattern of NMS 673 is less unusual (Figs. 4I and 4J). The first vertebral is wide and trapezoidal. Vertebrals 2–4 are wide, hexagonal elements. Laterally, vertebrals 2–3 extend slightly less than the mid-length of the costals, which is slightly less than in MHNN FOS 977. In NMS 101 and NMS 673, the sulcus between the fifth vertebral and the twelfth marginals is located just posterior to the suture between the second suprapygal and the pygal, whereas the sulcus is positioned around the mid-length of the pygal in MHNN FOS 977. Finally, the two Solothurn specimens are unique in having a first interpleural sulcus reaching the fourth marginal on the third peripheral, instead of the fourth as in most turtle, including the holotype of C. jaccardi (Fig. 4).

MHNN FOS 977, NMS 101, and NMS 673 also exhibit differences regarding their plastral morphology. The plastron is proportionally shorter in the Solothurn specimens (about 70% of the carapace length, as opposed to 78% in MHNN FOS 977; see Table 2). Their entoplastron is larger. In the holotype, the central plastral fontanelle is formed equally by the hyo- and hypoplastra, whereas in the Solothurn specimens it is formed mostly or entirely by the hypoplastra. The central plastral fontanelle is rounded in MHNN FOS 977 and NMS 101, but it is oval and narrow in NMS 673.

The aforementioned differences can be diversely interpreted and may ultimately warrant the placement of the Solothurn specimens in a different species. However, intraspecific variability (notably sexual dimorphism in the case of the variation of the relative plastral length), ontogenetic stage (NMS 101 and NMS 673 are about 15% larger than the holotype specimen), and stratigraphic age (MHNN FOS 977 is possibly slightly younger than the Solothurn specimens) may also explain at least part of these differences. In order to avoid the unnecessary creation of a new species, we still tentatively refer NMS 101 and NMS 673 to C. jaccardi. Hopefully, new discoveries will eventually shed light on this particular question. In the meantime, comparisons should be made primarily with MHNN FOS 977, the holotype of C. jaccardi.

Lapparent de Broin, Lange-Badré & Dutrieux (1996) proposed a number of relative proportions of various shell measurements in order notably to discriminate between the different Plesiochelys and Craspedochelys species (carapace length/width ratio, ratio between the length of the second intercostal sulcus and the width of the third vertebral, ratio between the length of the posterior plastral lobe and the length of the bridge, posterior plastral lobe length/width ratio, ratio between the length of the bridge and the length of the carapace, ratio between the length of the post-xiphiplastral space and the length of the carapace). However, many of these proportions are not discriminative and the others are too much influenced by postmortem deformation. In the course of the present study, we have also been looking for ratios that would allow to discriminate between the species at hand. As discussed above the ratio between the length of the plastron and the length of the carapace clearly differentiate C. jaccardi from P. etalloni (Table 2). For the other ratios, we have focused on individual bones whose measurements are not extensively affected by postmortem deformation. The length/width ratios of the hyoplastron and xiphiplastron discriminate between C. jaccardi and P. etalloni (Tables 4 and 5). In C. jaccardi, the hyoplastron is wider than long and the xiphiplastron about as wide as long, whereas the hyoplastron and xiphiplastron are both longer than wide in P. etalloni. The ratio between the length of the carapace and the length of the fourth costal reveals that the shell is proportionally wider in C. jaccardi than in P. etalloni and C. picteti (Table 6). Finally, the length/width ratio of the fourth costal is probably the most interesting feature, because it clearly allows to discriminate between the three aforementioned species. This ratio is high in C. jaccardi, slightly lower in C. picteti, and much lower in P. etalloni (Table 3). This is also clearly visible directly on the specimens where costals 2–6 seem thinner and elongate in C. jaccardi (Fig. 4), whereas they are wider and shorter in P. etalloni (Figs. 2 and 8). It is also interesting to note that measurements taken from MHNN FOS 977, NMS 101, and NMS 673 are generally congruent, which suggests that these specimens truly belong to a single species.

Table 5 Comparison of the length/with ratio of the xiphiplastron in selected specimens referred to P. etalloni, C. picteti, and C. jaccardi.

	Xiphiplastron
length (mm)	Xiphiplastron
width (mm)	Ratio	
Plesiochelys etalloni				
NMS 59	89	66	1.35	
NMS 78	84a	66	1.27	
NMS 79	81	59	1.37	
NMS 116	–	–	–	
NMS 669	79	68	1.16	
NMS 675	84	66b	1.27	
MAJ 2005-11-1	90c	68c	1.32	
Craspedochelys jaccardi				
NMS 101	62d	70	0.89	
NMS 102a	–	–	–	
NMS 612	–	–	–	
NMS 673	66	64	1.03	
MHNN FOS 977	55	55	1.00	
Craspedochelys picteti				
NMS 608	–	–	–	
Notes.

a Length incomplete.

b Width incomplete.

c From 3D surface mesh (see Anquetin, Deschamps & Claude, 2014).

d Length missing about 10 mm.

Table 6 Comparison of the ratio between the length of the carapace and the length of the fourth costal in selected specimens referred to P. etalloni, C. picteti, and C. jaccardi.

	Carapace
length (mm)	Costal 4
length (mm)	Ratio	
Plesiochelys etalloni				
NMS 59	474	176	2.69	
NMS 78	361a	152	2.38	
NMS 79	–	163	–	
NMS 116	–	–	–	
NMS 669	410	160	2.56	
NMS 675	445	178	2.50	
MAJ 2005-11-1	471	–	–	
Craspedochelys jaccardi				
NMS 101	413	188	2.20	
NMS 102a	363	164	2.21	
NMS 612	–	155	–	
NMS 673	411	181	2.27	
MHNN FOS 977	365	–	–	
Craspedochelys picteti				
NMS 608	540	217	2.49	
Notes.

a Carapace missing about 20 mm.

Tropidemys langii

In the Jura Mountains, the shell of Tropidemys langii is known from 19 specimens (out of which five are relatively complete) from the localities of Solothurn and Porrentruy, Switzerland (Fig. 1). In addition to this material, a partial carapace is also known from the site of Sainte-Croix (Canton of Vaud, Switzerland), but this specimen is supposed to have been found in Valanginian (Early Cretaceous) deposits. The material from Porrentruy is particularly important because it provides additional information regarding the plastron and limb bones (humerus and femur) of this species. All this material has been recently revised and described by Püntener et al. (2014).

The carapace of Tr. langii is tectiform in the posterior part. Its outline varies from oval to roundish. The nuchal is relatively variable in Tr. langii. The nuchal notch can be more or less pronounced, and is even absent in some individuals. MJSN VTT006-563 exhibits a pair of small supernumerary bones on the anterolateral edges of the nuchal, which changes its usually trapezoidal outline (Püntener et al., 2014: Fig. 8). Tropidemys langii is mainly characterized by the presence of thick, wide, hexagonal, and keeled neurals. Although the angle formed by the keel and the geometry of the neurals are subjected to some intraspecific variation (see Püntener et al., 2014: Table 1), these characters clearly distinguish Tr. langii from other Late Jurassic turtles, including Plesiochelys etalloni, Craspedochelys picteti, Craspedochelys jaccardi, Thalassemys hugii, and ‘Thalassemys’ moseri. The midline keel is barely noticeable on the anterior neurals, then it becomes progressively more pronounced posteriorly before subsiding on the suprapygals. It nevertheless reaches as far as the pygal posteriorly. The pygal region itself is relatively poorly known. There are usually two suprapygals that vary in shape among specimens. In some specimens, there seems to be an intermediate element between the eighth neural and the first suprapygal (Fig. 5), as in P. etalloni, C. picteti, C. jaccardi, and Th. hugii. The pygal is a wide and rectangular element. There are eight pairs of costals. It is remarkable that costals 1 and 2 curve strongly anteriorly in their distal parts, whereas costals 4–8 curve posteriorly. The third costal is straight and widens distally, compensating for the diverging curvature of costals 2 and 4.

All sufficiently preserved specimens of Tr. langii have three cervical scales, but they vary in shape and proportion. Tropidemys langii is characterized by its very narrow vertebral scales. The intervertebral sulci are usually convex anteriorly in the midline. The third intervertebral sulcus is variably located on the fifth or the sixth neural (Püntener et al., 2014: Table 1). In contrast to P. etalloni, C. picteti, C. jaccardi, and Th. hugii, in which the fourth intervertebral sulcus is usually located posterior to the eighth neural on the intermediate element, if present, or on the first suprapygal, this sulcus extends medially on the eighth neural in Tr. langii. Furthermore, the arrangement of vertebral scales distinguishes Tr. langii from Tr. seebachi Portis, 1878. This species is only known from the Kimmeridgian of Hannover, Germany, and is characterized by the presence of up to eight vertebral scales and an additional row of paired scales intercalated between the vertebrals and pleurals (Karl, Gröning & Brauckmann, 2012). The pleural scales are very wide in Tr. langii. The interpleural sulci are usually located on the posterior part of costals 2, 4, and 6, but the first intercostal sulcus may extend onto the third costal (e.g., MJSN VTT006-253 and NMS 15; see Fig. 5). MJSN VTT006-176 exhibits paired supernumerary pleural scales immediately lateral to the first vertebral (Püntener et al., 2014: Fig. 4B). The marginals of Tr. langii are generally rectangular in outline and about twice as long as wide. The fourth and fifth marginals may extend slightly onto the costals in some specimens (e.g., MJSN VTT006-253 and MJSN VTT006-563).

The plastral anatomy of Tr. langii was poorly known until Püntener et al. (2014) described some articulated material from Porrentruy. The connection between the carapace and plastron is relatively strong, as indicated by the extensive attachment sites for the plastral buttresses on the ventral surface of the first and fifth costals. The epi-, ento-, and xiphiplastron of Tr. langii are unknown. Similar to the condition observed in C. jaccardi (see Table 4), the hyoplastron of Tr. langii varies in proportion from about as wide as long (MJSN VTT006-290) to wider than long (MJSN VTT006-563). There is a central plastral fontanelle mainly formed by the hyoplastra (e.g., MJSN VTT006-290 and MJSN VTT006-563), but it is possible that the central plastral fontanelle was reduced or absent in some individuals, as suggested by MJSN VTT006-52 (Püntener et al., 2014).

As in C. jaccardi, the humeral scale is significantly longer than the pectoral scale. Similar to the condition observed on the carapace, the arrangement of plastral scales exhibits a certain amount of variability, such as the presence of supernumerary scales. For example, in MJSN VTT006-563, a small triangular scale is intercalated between the hyoplastra (Püntener et al., 2014: Fig. 12B). A similar supernumerary scale is known in MAJ 2005-11-1, the holotype of Plesiochelys etalloni (Fig. 2). Based on the available material, the anal scale was probably restricted to the xiphiplastron. There were apparently four inframarginals on each side, the second inframarginal being the longest in the series.

Thalassemys hugii

Bräm (1965) listed 15 specimens of Thalassemys hugii in the historic Solothurn collection. At the beginning of the 1990s, the Geological Institute of Bern, Switzerland, collected additional turtle remains from the locality of St Niklaus (Meyer & Thüring, 2009; and references therein). This rich material is now housed in the NMS. If most of this material remains undetermined up until today, we have been able to identify a few specimens as Th. hugii. However, a detailed review of all the Solothurn material assignable to Th. hugii goes beyond the scope of the present study, and we will simply provide important additional information on the plastral morphology of this species.

Bräm (1965) described the carapace of Th. hugii as heart-shaped, but most peripherals are missing in the holotype (NMS 1; Figs. 6A–6D). Lapparent de Broin, Lange-Badré & Dutrieux (1996) revealed that posterior peripherals were actually relatively wide and that the carapace was oval. The nuchal is a broad, trapezoidal element, much similar to that of Plesiochelys and Craspedochelys, but without nuchal notch. There are eight neurals. The first neural is quadrangular and notably broadened anteriorly. Neural 2–6 are elongate, hexagonal elements with shorter sides facing anteriorly. In NMS 1, the sixth neural is subdivided into two elements, but this is not interpreted as having any systematic value. The seventh and eighth neurals are shorter, hexagonal elements. Historically, authors have described three suprapygals in Th. hugii (Rütimeyer, 1873; Bräm, 1965; Lapparent de Broin, Lange-Badré & Dutrieux, 1996). Comparisons suggest that the arrow-shaped element located directly posterior to the eighth neural in NMS 1 may actually correspond to the ‘intermediate’ element described in Plesiochelys etalloni, Craspedochelys picteti, and Craspedochelys jaccardi (see above). The ventral aspect of this element indicates that it was articulated to the vertebral series, but only for the anterior half of its length. As discussed above for P. etalloni, identifying this element is rather difficult. Posterior to this arrowhead-shaped element, there are two suprapygals. The pygal is not preserved in NMS 1. There are eight pairs of costals. Costals 1 and 2 are sutured to peripherals 1–3 in adult individuals. Small costo-peripheral fontanelles are retained between remaining costals and peripherals.

Bräm (1965) and Lapparent de Broin, Lange-Badré & Dutrieux (1996) described only one cervical scale in Th. hugii, but examination of the type specimen suggests that three may have been present. A more detailed review of the available material would be necessary in order to determinate the number of cervical scales in this species. The first vertebral is trapezoidal and greatly enlarged anteriorly. Posteriorly, its width is similar to that of the nuchal, but the first vertebral reaches the middle of the second marginal scale anterolaterally. Vertebrals 2–4 are significantly narrower than the same elements in P. etalloni, C. picteti, and C. jaccardi. Vertebral 2 is the narrowest and shortest of these three scales, whereas vertebral 4 is the widest and longest. The outlines of vertebrals 2–4 are characteristic. Their anterior and posterior borders are mostly straight and transverse. Their anterolateral borders are always slightly concave laterally, whereas their posterolateral borders are usually straight. These anterolateral and posterolateral borders are usually of about the same length for a given vertebral scale. As a consequence of the reduced width of the vertebrals, pleurals 1–3 are clearly wider than long.

The reconstruction of the plastron proposed by Bräm (1965: Fig. 30) is mainly based on the plastron of the holotype (NMS 1), which is poorly preserved and gives a misleading image of the true plastral morphology of Th. hugii (Figs. 6C and 6D). Referred specimens, such as NMS 20, NMS 593, and NMS 22325, provide important indications (Fig. 9). The central plastral fontanelle is not as extensive as to prevent a median contact of the hypoplastra, as suggested by Bräm’s (1965) reconstruction. In contrast, the hypoplastra do meet posteriorly for about half of their length along a strongly interdigitating contact. Behind this contact there is a small xiphiplastral fontanelle that prevents the xiphiplastra from meeting anteriorly. More posteriorly, the xiphiplastra meet along an interdigitating contact. However, the most important characteristic revealed by specimens NMS 20, NMS 593, and NMS 22325 is the definitive presence of a lateral plastral fontanelle in Th. hugii (Fig. 9). Based mainly on NMS 1, Bräm (1965) concluded that a lateral plastral fontanelle was absent in Th. hugii, which allowed to differentiate this taxon from eurysternids like Eurysternum (see ‘Eurysternum’ ignoratum, above). However, Bräm (1965) overlooked the fact that a lateral plastral fontanelle is clearly present notably in NMS 20 and NMS 593. NMS 22325, a large right hyoplastron from St Niklaus (collected during the 1990s excavations by the Geological Institute of Bern, Switzerland) pertaining to a specimen that was only slightly smaller than the holotype, also indubitably shows a lateral plastral fontanelle. The presence of a lateral plastral fontanelle in Th. hugii calls into question the traditional diagnoses of the Thalassemydidae and Eurysternidae.

Figure 9 The plastral morphology of Thalassemys hugii.

NMS 593: (A) photograph of the right hyo- and hypoplastron in ventral view; (B) interpretative drawing of the right hyo- and hypoplastron in ventral view; (C) photograph of the right hyo- and hypoplastron in visceral view; (D) interpretative drawing of the right hyo- and hypoplastron in visceral view. NMS 22325: (E) photograph of the right hyoplastron in ventral view; (F) interpretative drawing of the right hyoplastron in ventral view; (G) photograph of the right hyoplastron in visceral view; (H) interpretative drawing of the right hyoplastron in visceral view. NMS 37251: (I) photograph of the shell in ventral view; (J) interpretative drawing of the shell in ventral view. Bones are white; stripped lines indicate internal bone layers; green solid lines indicate scales sulci; matrix is gray. Abbreviations: ax, axillary buttress; cax, contact for axillary buttress; co, costal; hyo, hyoplastron; hypo, hypoplastron; in, inguinal buttress; lpf, lateral plastral fontanelle; xi, xiphiplastron.

During our review of the material, we have also identified two additional characters that allow to differentiate Th. hugii from other Solothurn turtles. The first of these characters is the presence of well-developed linear striations perpendicular to sutures between most shell elements, somewhat recalling the condition known in the Early Cretaceous Pleurosternon bullockii (e.g., Milner, 2004). These striations are clearly visible notably in NMS 1 (see Rütimeyer, 1873: plate 1; Bräm, 1965: plate 7, Fig. 4; Fig. 6), NMS 9, and NMS 22326-22327 (costals associated with the hyoplastron NMS 22325). They are also present in several specimens previously referred to E. ignoratum (see above): NMS 5, NMS 124, and NMS 412. The second character is the presence of a strong axillary buttress that is articulated over a large area on the ventral surface of the distal part of the first costal, as seen in NMS 1, NMS 412, and NMS 37251. The inguinal buttress is also relatively massive, although less so than the axillary buttress.

Bräm (1965) designated NMS 5 as the holotype of Eurysternum ignoratum and further referred NMS 124 (but see Plesiochelys langii, above) and NMS 412 to this species. However, these specimens are indiscernible from other specimens referred to Th. hugii (Fig. 6): e.g., vertebral scales with similar outlines and proportions (e.g., second vertebral length/width ratio of about 69% and 72% in NMS 1 and NMS 412, respectively); presence of clearly visible linear striations perpendicular to sutures between most shell elements (present in NMS 5, NMS 124, and NMS 412); presence of a strong attachment site for a large axillary buttress on the ventral surface of the distal part of the first costal (visible only in NMS 412). As in Th. hugii (see above; not Bräm, 1965), the plastron of NMS 5 has lateral plastral fontanelles and a central plastral fontanelle closed anteriorly by a median, interdigitating contact of the hyoplastra. A preliminary comparison of the girdle elements (notably the scapula and pubis) of NMS 5 (holotype of E. ignoratum), NMS 1 (holotype of Th. hugii), and NMS 9 (a specimen referred to Th. hugii) also reveals a very close morphology. Therefore, E. ignoratum is interpreted herein as a subjective synonym of Th. hugii.

‘Thalassemys’ moseri

Bräm (1965) typified ‘Thalassemys’ moseri based on a partial carapace and plastron (NMS 618; Figs. 7A–7D). He also referred three additional specimens to this species: a partial carapace (NMS 62; Figs. 7E and 7F), an isolated right hyoplastron (NMS 64), and an isolated left hyoplastron (NMS 111). As already noted by Bräm (1965) and Lapparent de Broin, Lange-Badré & Dutrieux (1996), the carapace of ‘Th.’ moseri is superficially similar to that of Plesiochelys etalloni: large trapezoidal nuchal with a broad and shallow nuchal notch; neurals elongated; three cervical scales; vertebrals wide and hexagonal with slightly sinuous sulci. However, ‘Th.’ moseri is characterized by the retention of costo-peripheral fontanelles in adults. NMS 618 and NMS 62 would have had an approximate carapace length of 400 mm. Specimens of similar size referred to P. etalloni are common in Solothurn (e.g., NMS 78 and NMS 107), but all have a completely ossified carapace. Furthermore, NMS 606, a juvenile P. etalloni with a carapace length of about 200 mm, also have an entirely ossified carapace. The retention of costo-peripheral fontanelles in adults is therefore a diagnostic feature of ‘Th.’ moseri. Close examination of NMS 618 and NMS 62 also reveals that their costals are very thin distally. This is clearly different from the condition known in Th. hugii, in which the costals remain relatively thick distally. Hence, the costals taper progressively distally in ‘Th.’ moseri, whereas their distal end is proportionally thicker and blunt in Th. hugii.

The plastron of ‘Th.’ moseri is best known from the holotype specimen (NMS 618). It is characterized by the presence of a central plastral fontanelle that is proportionally larger than that of P. etalloni or C. jaccardi. In contrast to Th. hugii (see above; not Bräm, 1965), the central plastral fontanelle is closed anteriorly and posteriorly by tight sutural contacts of the hyo- and hypoplastra, respectively. There is no lateral plastral fontanelle. Bräm (1965) noted that the epi- and entoplastron were not suturally connected to the hyoplastron. This reminds the condition in Th. hugii, but clearly departs from the strong sutural contact observed in P. etalloni and C. jaccardi. Finally, as suggested by Bräm (1965), there may have been a small xiphiplastral notch posteriorly.

Rieppel (1980) described a skull and associated, fragmentary shell remains (PMZH A/III 514) from the early Tithonian of La Morelière (Isle of Oléron, France) that he referred to ‘Th.’ moseri. Subsequent authors disagreed with this referral, considering that the specimen from La Morelière was a different taxon (Lapparent de Broin, Lange-Badré & Dutrieux, 1996; A Pérez-García, pers. comm., 2014). However, none of these authors studied the material first-hand. According to Rieppel’s (1980) conclusions, ‘Th.’ moseri is more closely related to Plesiochelys than to Thalassemys, but many features in the skull of ‘Th.’ moseri are plesiomorphic compared to the same features in P. etalloni and Portlandemys mcdowelli Gaffney, 1975a. Consequently, a referral of ‘Th.’ moseri to Plesiochelys does not seem appropriate. In the present study, we furthermore reveal that Th. hugii has a lateral plastral fontanelle, a feature absent in ‘Th.’ moseri. It therefore seems improbable that ‘Th.’ moseri is the closest relative of Th. hugii and its referral to Thalassemys does not seem appropriate either. In the current state of knowledge, the generic assignment of ‘Th.’ moseri remains uncertain. A thorough revision of the specimen described by Rieppel (1980) would certainly be an essential step toward a better understanding of this species, but ultimately more material is needed first to confirm or refute Rieppel’s (1980) identification, and second to gain insight into the morphology and relationships of this turtle.

Conclusions

Fifteen species of eucryptodires have historically been typified based on material from the Late Jurassic of the Jura Mountains. Bräm (1965) proposed the last systematic review of all the available material from Solothurn and still recognized nine out of these 15 species. Subsequent studies focused their attention mainly on the genera Plesiochelys and Craspedochelys, representing a total of five species according to Bräm’s (1965) taxonomy. Gaffney (1975a) united these five species into a single one (P. etalloni), whereas Lapparent de Broin, Lange-Badré & Dutrieux (1996) recognized four out of five species as valid (P. etalloni, P. solodurensis, C. picteti, and C. jaccardi).

The present study is the most complete taxonomic review of the Late Jurassic eucryptodiran turtles from the Jura Mountains since Bräm (1965). Its purpose was not only to reassess the taxonomy of these turtles, but also to reevaluate the available material in light of recent knowledge. We have not only considered the type specimens, but have also directly observed numerous referred specimens notably from the Solothurn collection (see Table S1). Out of the original 15 species, we recognize six as valid: Plesiochelys etalloni, Craspedochelys picteti, Craspedochelys jaccardi, Tropidemys langii, Thalassemys hugii, and ‘Thalassemys’ moseri. For the time being, these species are assigned to the traditional families Plesiochelyidae (P. etalloni, C. picteti, C. jaccardi, and Tr. langii) and Thalassemydidae (Th. hugii), although the proper definition of these taxa needs to be reconsidered in a phylogenetic context. The generic and suprageneric assignment of ‘Th.’ moseri remains conjectural. The presence of lateral plastral fontanelles in Th. hugii calls into question the traditional distinction between the Thalassemydidae and the Eurysternidae. Since Eurysternum ignoratum is considered a junior synonym of Th. hugii, the fossil record of eurysternids in the Late Jurassic of the Jura Mountains should be regarded as relatively sparse. Indeed, they are now only represented by a single skull of Solnhofia parsonsi from Solothurn (Gaffney, 1975b).

If the present taxonomic review represents a dramatic reduction in terms of number of species, the presence of six more or less closely related, relatively large coastal marine turtles in the same paleoenvironment is still remarkable. More than 60 fossil turtle taxa have been typified based on Late Jurassic European material. A global taxonomic revision of these turtles is long overdue. We are aware that the present study is only regional in scope and that some adjustments may be necessary in years to come following the revision of Late Jurassic turtles from other parts of Europe, notably in Germany, UK, France, Spain, and Portugal. The present paper will nonetheless serve as a base for future work on Late Jurassic European eucryptodires, notably for the study of the rich Kimmeridgian material unearthed by the PAL A16 team in the vicinity of Porrentruy, Switzerland. Institutional Abbreviations

MAJ Musée d’archéologie du Jura, Lons-le-Saunier, France

MH Naturhistorisches Museum, Basel, Switzerland

MHNN Muséum d’histoire naturelle, Neuchâtel, Switzerland

MJSN Musée jurassien des sciences naturelles, Porrentruy, Switzerland

NMS Naturmuseum Solothurn, Switzerland

PMZH Paläontologisches Institut und Museum, Universität Zürich, Switzerland

Supplemental Information

Table S1 List of specimens studied as part of the present work

Note that not all of these specimens are mentioned in the text, but all have been scrutinized as part of the present study. Note the new specimen numbers for the specimens housed in the Naturmuseum Solothurn (NMS).

Click here for additional data file.

We thank Silvan Thüring (NMS), Loïc Costeur (MH), Sylvie Deschamps (MAJ), and Christophe Dufour (MHNN) for providing access to specimens in their care. Thanks are extended to Yves Maître (PAL A16 team) for his assistance with Fig. 1. Comments from Walter Joyce, Adán Pérez-García, and an anonymous reviewer greatly improved the manuscript.

Additional Information and Declarations

Competing Interests

Author Contributions

Jérémy Anquetin is an Academic Editor for PeerJ.

Jérémy Anquetin conceived and designed the experiments, performed the experiments, analyzed the data, wrote the paper, prepared figures and/or tables, reviewed drafts of the paper.

Christian Püntener performed the experiments, analyzed the data, wrote the paper, prepared figures and/or tables, reviewed drafts of the paper.

Jean-Paul Billon-Bruyat analyzed the data, wrote the paper, reviewed drafts of the paper.

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
