# Peer review of "A taxonomic review of the Late Jurassic eucryptodiran turtles from the Jura Mountains (Switzerland and France)"

_PeerJ, doi:10.7717/peerj.369_

## Round 0.1 · original submission · Minor Revisions

I think that the requests of the Reviewers are reasonable and I encourage you to take them into consideration as you revise your manuscript.

·

Basic reporting

Dear Editors,

Review of Anquetin et al.: “A taxonomic review of the Late Jurassic turtles from the Jura Mountains (Switzerland and France).”

This manuscript is concerned with the taxonomic review of all turtle taxa named from the Jura Mountains of Switzerland and France. All in all, this is a really nice manuscript: the illustrations are well made, the English only demands minor corrections, and I have no difficulty replicating all scientific aspects. I greatly encourage its eventual publication.

I marked up with the manuscript with small comments highlighting occasional mistakes in syntax, spelling, and grammar and making suggestions on how to improve the manuscript. The most major point include:
1)
2) As I mentioned above, the figures are great, but all pictures and illustrations are much too small! Given that these is an e-Journal, there really is no need to safe with space or printing costs and I encourage the authors to liberally split the figures. Please keep in mind that the resolution of the figures is fixed (300dpi) and that much information will be lost through unnecessary pixilation, especially in the photos, if the figures are kept small.
3) The authors promise a complete review of turtles from the Jura, but fail to list two taxa. At the very least, the authors need to include Helemys serrata, as this is typified based on material from Switzerland. The authors should furthermore think about including Solnhofia parsonsi, because material from the Jura has been referred to this taxon. Finally, the authors should highlight the fact that their review, once again, is only regional. Who knows what the result would be if all Jurassic turtles were included from Europe!
4) The authors have a section “referred material” for all taxa, but this only makes sense for valid taxa, as all material referred to an invalid taxon is actually referred to its senior synonym.
5) There similarly is no point of having a section “synonymies” for invalid taxa.
6) Finally, the authors need to keep in mind that all species should be correctly assigned in the Systematic Paleontology. So, whereas a note should be added to Eurysternum, the taxon “Eurysternum ignoratum” needs to be listed under Thalassemys, because it is now referred to that taxon through synonymy with T. hugii.
7)
All of these points are really minor and I am certain that the authors will agree. I therefore think that the authors should be able to address all of these points quickly and I do not see any need for an additional round of reviews.

The authors are welcome to know my identity.

Sincerely,

Walter Joyce

Experimental design

see above

Validity of the findings

see above

Additional comments

see above

Reviewer 2 ·

Basic reporting

No Comments

Experimental design

No Comments

Validity of the findings

The manuscript „A taxonomic review of the Late Jurassic turtles from the Jura Mountains (Switzerland and France)” provides a detailed, revised taxonomy of the Late Jurassic (mainly Kimmeridgian) turtles from the Jura Mountains in France and Switzerland and represents an important first step in a revision of the taxonomy and diversity of Late Jurassic turtles from Central Europe. The manuscript adds to our understanding of the diversity of turtles in the Jurassic of Europe as most of the earliest named species that were traditionally referred to Plesiochelyidae, Thalassemydidae and Eurysternidae are from this area and have long been in need of a taxonomic reevaluation. Therefore publication is strongly recommended after some changes to the manuscript have been addressed.

My main concern is that I wonder why the authors did not include a phylogeny of the revised turtle taxa. I am aware that taxonomic papers could go without a phylogeny and that the description of the individual taxa is thorough. However, a phylogeny including the new characters defining the taxonomy of the discussed turtle taxa from the turtle assemblage of the Jura Mountains (especially since these localities provided holotype material of so many traditional taxa) would provide a stronger basis for future studies of Late Jurassic taxa from Central Europe as the authors suggest in their conclusion.

Moreover, the “methodology” (line 1119) used by the authors is simply a descriptive discussion of the taxonomy and anatomy of each taxon. To test the validity and relationships of the taxa under study, a phylogenetic framework is crucial. I strongly urge the authors to consider testing their hypotheses incorporating them into a phylogeny (e.g. Püntener et al, 2014, using a phylogeny based on Joyce, 2007).

Additional comments

Minor editorial changes and comments for the text:

Line 226: “NMS 118, a large carapace missing both sides”. Please rephrase. What is missing?

Line 870: “ontogenetic development” Did the authors mean "ontogenetic stage" (as the different size of the mentioned specimens indicates) or is there more than one different ontogenetic development?

Line 931-934: there is a comma missing before “in which”.

Line 954: individuals not individual

·

Basic reporting

I considered the manuscript of Anquetin et al. as very interesting, and I think that, after considering my small suggestions, it should be published in this journal.

My suggestions are:

-Lines 165 to 166: Why not to delete that sentence? Several papers can be considered as “steps”.
-Line 187: Can you add some taxa as examples?
-Line 211: Why not to add here the diagnosis (indicating something like: taken form Anquetin et al, 2014)?. This would help the reader.
-Lines 254 and 260: Why you need to propose a lectotype and paralectotypes of a no valid taxon? I think this is not necessary. If you like to propose it, you need to justify it because this is a necessary requirement of the International Code of Biological Nomenclature.
-Line 330: Please, delete the character “and a hyoplastron only slightly longer than wide” because it is indicated, in the same diagnosis, as a specific character (see that the last character of the diagnosis is the same), and of a character shared with all the members of the genus. See also the description of this character in the diagnosis of C. jaccardi (and check the description of this character in the diagnosis of both species).
-In the comparative diagnosis of both species of Craspedochelys most of the characters are compared, but not all of them. You need to compare all of them (or to delete the characters not known in one of them if this is the case).
-Line 466: Why not to add here the diagnosis?
-Lines 511-512: Delete “great anterior widening of first neural”. This character is variable in Thalassemys. In fact, many of the specimens of Plesiochelys represented in this paper have that the first neural wider than in some known specimens of Thalassemys.
-Line 567-597: I suggest deleting of this part the specimen from the Oléron Island: As the authors indicate, its attribution to the same species has to be checked.
-Line 712: Please, could you indicate what is that “common pattern”?
-Line 935: You can indicate “other valid species” but not “the other valid species”. The validity of other classical species is justified in other accepted paper, but not in press for the moment.

Experimental design

All is OK

Validity of the findings

All is OK

---

## Round 0.2 · accepted · Accept

You have done a nice job responding to the reviewer suggestions and your manuscript will make a valuable contribution to the literature.